**Modeling impacts of dust mineralogy on fast climate response**
Qianqian Song[1], Paul Ginoux[2], María Gonçalves Ageitos[3,4], Ron L. Miller[5,6], Vincenzo Obiso[5],
Carlos Pérez García-Pando[4,7]
[1] Atmospheric and Oceanic Sciences Program, Princeton University, Princeton, NJ, USA
[2] NOAA Geophysical Fluid Dynamics Laboratory, Princeton, NJ, USA
[3] Projects and Construction Engineering Department. Universitat Politècnica de Catalunya-Barcelona
TECH, Terrassa, Spain
[4] Barcelona Supercomputing Center, Barcelona, Spain
[5] NASA Goddard Institute for Space Studies, New York, NY, USA
[6] Department of Applied Physics and Applied Mathematics, Columbia University, New York, NY, USA
[7] ICREA, Catalan Institution for Research and Advanced Studies, Barcelona, Spain
Correspondence: Qianqian Song (qs7080@princeton.edu)

# Abstract

Mineralogical composition drives dust impacts on Earth's climate systems. However, most climate models still use homogeneous dust, without accounting for the temporal and spatial variation in mineralogy. To quantify the radiative impact of resolving dust mineralogy on Earth's climate, we implement and simulate the distribution of dust minerals (i.e., illite, kaolinite, smectite, hematite, calcite, feldspar, quartz, and gypsum) from Claquin et al. (1999) (C1999) and activate their interaction with radiation in the GFDL AM4.0 model. Resolving mineralogy reduces dust absorption compared to the homogeneous dust used in the standard GFDL AM4.0 model that assumes a globally uniform hematite volume content of 2.7% (HD27). The reduction in dust absorption results in improved agreement with observation-based single scattering albedo (SSA), radiative fluxes from CERES (the Clouds and the Earth's Radiant Energy System), and land surface temperature from CRU (Climatic Research Unit), compared to the baseline HD27 model version. It also results in distinct radiative impacts on Earth's climate over North Africa. Over the 19-year (from 2001 to 2019) modeled period during JJA (June-July-August), the reduction in dust absorption in AM4.0 leads to a reduction of over 50% in net downward radiation across the Sahara and approximately 20% over the Sahel at top of atmosphere (TOA) compared to the baseline HD27 model version. The reduced dust absorption weakens the atmospheric warming effect of dust aerosols and leads to an alteration in land surface temperature, resulting in a decrease of 0.66 K over the Sahara and an increase of 0.7 K over the Sahel. The less warming in the atmosphere suppresses ascent and weakens the monsoon inflow from the Gulf of Guinea. This brings less moisture to the Sahel, which combined with decreased ascent induces a reduction of precipitation. To isolate the effect of reduced absorption, compared to resolving spatial and temporal mineralogy, we carry out a simulation where the hematite volume content of homogeneous dust is reduced from 2.7% to 0.9% (HD09). The dust absorption (e.g., single scattering albedo) of HD09 is comparable to that of the mineralogically speciated model on a global mean scale, albeit with a lower spatial variation that arises solely from particle size. Comparison of the two models indicates that the spatial inhomogeneity in dust absorption resulting from resolving mineralogy does not have significant impacts on Earth's radiation and climate, provided there is a similar level of dust absorption on a global mean scale before and after resolving dust mineralogy. However, uncertainties related to emission and distribution of minerals may blur the advantages of resolving minerals to study their impact on radiation, cloud properties, ocean biogeochemistry, air quality,

and photochemistry. On the other hand, lumping together clay minerals (i.e., illite, kaolinite, and
smectite), but excluding externally mixed hematite and gypsum, appears to provide both
computational efficiency and relative accuracy. Nevertheless, for specific research, it may be
necessary to fully resolve mineralogy to achieve accuracy.

# 1  Introduction

Soil dust aerosols emitted from erodible land surfaces, hereafter referred to as dust, are the most
abundant aerosol component in the atmosphere in terms of dry mass. Dust has significant impacts on
the Earth's climate systems (atmosphere, ocean, cryosphere) due to its interaction with terrestrial and
solar radiation (Sokolik and Toon, 1999), cloud microphysics (Guo et al., 2021), tropospheric
chemistry (Bian and Zender, 2003; Paulot et al., 2016), and oceanic and terrestrial
biogeochemistry(Mahowald, 2011; Evans et al., 2019; Dunne et al., 2020). In addition, dust particles
deposited on snow and ice decrease surface reflectivity and accelerate snowmelt (Skiles et al., 2018;
Réveillet et al., 2022). Dust can influence Earth's radiative energy budget through different pathways;
among them: 1) directly by interacting with both solar and terrestrial radiation (i.e., direct radiative
effects, hereafter referred to as DRE) 2) by radiatively influencing the thermal dynamical structure of
atmosphere and thereby clouds (i.e., semi-direct radiative effect) and 3) indirectly by altering cloud
reflectivity (cloud albedo effect) and lifetime (cloud lifetime effect). Unfortunately, the quantitative
estimate of dust DRE at the top of atmosphere (TOA) is largely uncertain (Claquin et al., 1998; Miller
et al., 2014; Kok et al., 2017; Song et al., 2022). A significant part of this uncertainty has been attributed
to neglecting variations in dust mineralogical composition and its evolution during transport (Li et al.,

65 2021).

The magnitude of dust impacts on the Earth's climate systems depends on its mineralogical
composition, as has been shown in multiple studies. In the ShortWave (SW), dust absorption depends
on the iron oxides content. Sokolik and Toon (1999) suggested that a small amount of iron oxides
internally mixed with less absorptive minerals is able to reverse the sign of $DRE_{SW}$ at TOA from
negative (cooling effect) to positive (warming effect). Multiple studies have confirmed the importance
of iron oxides to the dust $DRE_{SW}$ (Balkanski et al., 2007; Li et al., 2021; Obiso et al., 2023). In
LongWave (LW) spectrum, absorption and $DRE_{LW}$ depend on the abundance of quartz, calcite, and
clays in coarse and super-coarse modes (Di Biagio et al., 2017; Sokolik and Toon, 1999). As a result,
resolving dust mineralogy allows to better understand the impact of dust DRE, such as the fast response
of the land surface temperature, as opposed to the slow response of sea surface temperature that will
not be studied here. This fast temperature response will affect precipitation, and atmospheric
circulation (Ming et al., 2010; Persad et al., 2014).
In addition, resolving dust mineralogy is also crucial for studying heterogeneous reactions of acid gases
with dust aerosols. For example, the uptake of $HNO_3$, $NO_3^-$, $N_2O_5$ on dust particles is suggested to be
limited by alkalinity that comprises calcium and magnesium carbonates (Song and Carmichael, 2001;
Paulot et al., 2016). These reactions will modify the composition of dust particles and subsequently
changing their hygroscopicity, cloud condensation nucleation (CCN), and ice nucleation activities
(Kelly et al., 2007), and thereby further affecting precipitation (Rosenfeld et al., 2001). Moreover,
heterogeneous reactions with mineral dust could significantly affect tropospheric photochemical
oxidation cycles, causing up to 10% reduction in O3 concentrations in dust source regions and nearby
(Dentener et al., 1996). Among the different minerals, K-Feldspar appears to dominate ice nucleation,
despite being a minor component of aeolian dust (Atkinson et al., 2013; Harrison et al., 2019), although
other minerals such as quartz may also contribute (Chatziparaschos et al., 2023). The key factor
controlling the production and removal of pollutants and the damages by acid rain is the pH of
raindrops, which has been observed to increase due to its dependency on Ca-rich dust (Grider et al.,

91  2023).

Despite the potential importance of resolving dust mineralogy in various aspects, current climate
models tend to use a fixed mineralogy without considering the temporal and spatial variations in
dust mineralogical composition. To test the importance of resolving dust mineralogy on the fast
climate response (e.g., surface temperature response over land, atmospheric circulation and
precipitation response) through its interactions with SW and LW radiation (i.e., through dust DRE),
dust mineralogy has been implemented and simulated in the GFDL AM4.0 model (Zhao et al.,
2018a, b), including its on-line interactions with radiation. Following the pioneering work of
Claquin et al. (1999) (C1999), we consider the emission, transport and interactions with radiation
and deposition of eight minerals: illite, kaolinite, smectite, hematite, calcite, feldspar, quartz and
gypsum. Following the recent launch of the Earth Surface Mineral Dust Source Investigation
(EMIT) instrument specifically designed to retrieve global distribution of dust mineralogy over
dust sources (Green et al., 2020), there have been coordinated efforts to represent dust mineralogy
and investigate DRE of mineral-speciated dust in climate models, in particular in Li et al. (2021),
Gonçalves Ageitos et al. (2023), and Obiso et al. (2023). However, to the best of our knowledge,
there have been no studies investigating the fast climate impact of dust while accounting for its
mineral speciation. Our work contributes to these efforts by incorporating dust mineralogy into the
GFDL models, and it is distinguished by extending its investigation to the fast climate response of
mineral-speciated dust. The impacts of dust mineralogy on other aspects, such as sea surface
temperature and slow climate response, heterogeneous reactions, and ice nucleation ability, will
be examined in future studies.
Section 2 provides the description of the GFDL AM4.0 model and dust mineralogy
implementation. Section 3 describes our experimental design. In Section 4, we calculate mineral
optical properties, activate the interaction of minerals with radiation in GFDL AM4.0 and compare
modeled dust optical properties with observations. Section 5 presents the impacts of resolving dust
mineralogy on Earth's radiation and climate with a focus on the North Africa, as well as their
evaluations. In Section 6, we investigate the influences of reducing the number of mineral tracers.
Section 7 provides a summary of the study along with the main conclusions.

## 2   Model and Data

### 2.1   Model description

We conduct a series of experiments with GFDL AM4.0 (Zhao et al., 2018a, b) over the period
2001-2019. These experiments use the AMIP protocol, where sea surface temperature (SST) and
sea-ice are imposed based upon average monthly observations (see Gates, 1992 for details) .
Observed gridded SST and sea-ice concentration boundary conditions are from the reconstructions
of Taylor et al., (2000). Historical reconstructions of monthly solar spectral irradiances are from
Matthes et al., (2017). For radiation calculations, global monthly mean concentrations of
greenhouse gases (GHGs), including nitrous oxide ($N_2O$), and ozone-depleting substances (ODSs,
including CFC-11, CFC-12, CFC-113, and HCFC-22) are specified from Meinshausen et al.,
(2017). The solar irradiances and GHG databases are standard for CMIP6. Longwave (LW)
scattering of aerosols is not accounted for in the model.
In AM4.0, dust emission is calculated interactively following the parameterization of Ginoux et al.
(2001) with a threshold of wind erosion and global scaling factor of 3.5 $m\ s^{-1}$ and 0.2 $\mu g\ s^2\ m^{-5}$,
respectively. Dust size is represented by five bins with diameter ranging from 0.2 $\mu m$ to $20\mu m$
(bin1: 0.2 - 2 $\mu m$, bin2: 2 - 3.6 $\mu m$, bin3: 3.6 - 6 $\mu m$, bin4: 6 - 12 $\mu m$, bin5: 12 - 20 $\mu m$). The
corresponding source fractions have been updated from 0.1, 0.225, 0.225, 0.225 and 0.225 to
values of 0.04, 0.14, 0.19, 0.49, and 0.14 for the five bins. These updated source functions allocate
more fraction to coarser size bins, following the suggested Brittle Fragmentation Theory (BFT) as
proposed by Kok et al. (2011). Dust mineral composition in the standard AM4.0 is considered as
uniform, with no temporal and spatial variations; in other words, dust Refractive Index (RI) is
temporally and spatially homogeneous (case referred to as homogeneous dust hereafter). The dust
RI in the standard AM4.0 is taken from Balkanski et al., (2007), assuming a fixed hematite content
of 2.7% by volume (HD27), which was calculated for the internal mixture of hematite and five
other minerals (calcite, quartz, illite, kaolinite and montmorillonite) using the Maxwell Garnett
mixing rule (see details in Balkanski et al., 2007). The decision to fix the hematite content for dust
particles at 2.7% was made during the development of the previous GFDL Climate Model CM3
(Donner et al., 2011). This decision was prompted by the discovery that dust absorption was
unrealistically high (by a factor 3) in CM2 (Delworth et al., 2006) compared to AERONET
observations (Balkanski et al., 2007). In CM3, the conjunction of a sharp decrease of black carbon
(strong aerosol absorber) with a new emission inventory and the switch to more scattering dust
had a negative effect on precipitation bias, and late 20th century warming (see Donner et al., 2011
for details). To mitigate this bias, the selection of 2.7% hematite was adopted in CM3, as well as
in the subsequent GFDL models. The control run conducted with the homogeneous dust in the
standard AM4.0 model is labeled as HD27 as described in Table 2.
In addition, we conduct simulations assuming homogeneous dust with hematite content of 0.9%
by volume, with RI from Balkanski et al. (2007). Similar to HD27, this experiment, labeled as
HD09 in Table 2, does not account for the temporal and spatial variations in dust mineralogy.

## 2.2 Dust mineralogy implementation

Claquin et al. (1999) (C1999) is the earliest study providing a soil mineralogy map oriented toward
atmospheric and climate modelling. The soil map provides the mineral mass fractions present in
the clay and silt size ranges for eight different minerals, namely: illite, smectite, kaolinite, calcite,
quartz, feldspars, gypsum, and hematite. In this study, we implement the eight minerals from the
soil mineralogy map provided by C1999 in GFDL AM4.0 to resolve dust mineralogy. The soil
map is based on soil analyses that are usually done after wet sieving, which disperse mineral
aggregates into small particles. This dispersal is particularly relevant for the phyllosilicates,
typically found in the form of aggregates in soils. They are detected in the atmosphere with higher
proportions at coarser (silt) sizes than those reported in the soil maps (Perlwitz et al., 2015b; Perez
Garcia-Pando et al., 2016). These recent studies also show that the Brittle Fragmentation Theory
(BFT; Kok, 2011) represents a practical framework to generate the emitted particle size
distribution based on the dispersed soil PSD, which facilitates the utilization of soil mineralogy
maps. In our simulations, we employ BFT to reconstruct the mineral aggregates emitted from the
original undispersed soils, following the methods described in Gonçalves Ageitos et al., 2023. The
mass density of the eight minerals, along with a brief description of their importance to Earth's
climate, are listed in Table 1. The density of minerals impacts their settling velocity, which is
relevant to the removal of particles in the atmosphere. Goethite and hematite are the two major
types of iron oxides present in soils. Goethite is less absorptive than hematite and is not resolved
in C1999. So, iron oxides are represented by hematite in this study. Hematite has larger density
than other minerals, so that hematite deposits more quickly and is not able to be transported to
remote regions when not aggregated or internally mixed with lighter clay minerals. Moreover,
among the minerals considered here, hematite is the strongest absorber at ultraviolet (UV) and
visible wavelengths, while it does not have noticeable absorption at infrared wavelengths (IR)
compared to other minerals (Sokolik and Toon, 1999). As such, the correct representation of
hematite content in dust aerosols is critical in improving the representation of dust interaction with
SW radiation in climate models. All minerals are considered to be externally mixed, except for
iron oxides. A large part of the emitted flux of iron oxides is considered to be internally mixed
with other minerals, e.g., in the form of accretions in phyllosilicates, in line with observational
evidence and previous modeling studies (Kandler et al., 2009; Perlwitz et al., 2015a; Zhang et al.,
2015; Panta et al., 2023). As suggested by Gonçalves Ageitos et al. (2023), we define two different
types of tracers for the iron oxides: one set of tracers carries the mass of the hematite that
constitutes small accretions in clay minerals (i.e., internally mixed with clay minerals), are allowed
to be up to 5 % of the masses of their host minerals at emission (Perlwitz et al., 2015a; Gonçalves
Ageitos et al., 2023). Given the low fractional mass of hematite compared to their host minerals,
we assume that these accretions do not change the density of their host particles. These internally
mixed accretions form the largest fraction of the emitted hematite. Another smaller set of tracers
carries the mass of the remaining fraction of hematite, which is considered to be externally mixed
with the other minerals, including the internal mixtures of hematite with clay.
In addition to the similar roles of clay minerals in carrying iron oxides, the optical properties of
the three clay minerals are very similar, and the optical properties of their external mixture are
found to be almost identical to their internal mixture (see Section S1 in the Supplement). This
finding suggests the use of a single mineral species to represent all three clay minerals in their
interaction with radiation to reduce computational cost. Therefore, the optical properties of one
single mineral (clay433) are used to represent the optical properties of all three clay minerals. The
clay433 represents a mixed mineral comprising three clay minerals: illite, kaolinite and smectite,
with mass fraction of 40%, 30%, and 30%, respectively (see detailed descriptions in
Supplementary Section S1). This simplification streamlines the calculations of optical properties
for internal mixtures of hematite and the three clay minerals (illite, kaolinite, and smectite),
reducing it from an internal mixture of four minerals (hematite, illite, kaolinite and smectite) to an
internal mixture of two minerals (hematite and clay433).
The optical properties of the internal mixture of hematite and clay433 are calculated using three
mixing rules: volume weighted average (VOL-mixing), Maxwell-Garnett mixing rule (MG-
mixing) and Bruggeman mixing rule (BM-mixing). Generally, VOL-mixing is used for a quasi-
homogeneous mixture, that is when the components have similar refractive index. For cases
involving dominant homogeneous host with small inclusions of contrasting composition, MG-
mixing is appropriate. BM-mixing is suitable for mixtures that the inclusions virtually occupy the
entire volume of the particle, and the host disappears. The detailed discussion regarding the three
mixing rules and their applications can be found in Liu and Daum (2008) and Markel (2016). The
appropriate selection of mixing rules is important for the determination of the optical properties of
the mixtures. Therefore, we incorporate all three mixing rules in this study. These calculations are
performed for various volume mixing fractions of hematite with respect to clay433, to construct a
lookup table (LUT) for each mixing rule. The optical properties of each mineral as well as the
internal mixtures of hematite and clay433 are calculated offline using Mie code with a spherical
shape assumption. As all other minerals have similar SW absorption, internal or external mixing
does not change their absorption properties. So, we assume all other minerals to be externally
mixed. More details about optical properties of minerals will be discussed in Section 4.
Overall, we implement nine types of mineral tracers: seven non-hematite minerals along with
distinguished internal and external hematite, as listed in Table 1. Each type is distributed across
five size bins. As a result, 45 mineral tracers have been incorporated in AM4.0 to account for dust
mineralogy.
Table 1. The list of minerals considered in this study and their importance to Earth's climate. Mineral-
dependent mass densities are defined following Table 1 in Gonçalves Ageitos et al. (2023), in which the
references of mineral densities are listed.

| Minerals | Density (kg/m$^3$) | Importance |
|---|---|---|
| 1.    Hematite (int.) | 2570 | It is the strongest visible absorber. It is internally mixed with clay minerals when its mass fraction at emission < 5%. |
| 2.    Hematite (ext.) | 4770[1] | It is externally mixed for the part of emitted mass fraction > 5%. |
| Three clay minerals: 3.    Illite 4.    Smectite 5.    Kaolinite     Clay in BM-RT | 2570 2570 2630    2590 | They are the most abundant mineral components in clay-sized (diameter $< 2\mu m$) minerals. They are internally mixed with internal hematite.    The three clay minerals are lumped together as one mineral species 'clay' in the BM-RT experiment in Section 6. |
| 6.    Calcite | 2710 | It is important for chemistry. (e.g., heterogeneous reaction with acidic gases and formation of sulfate and nitrates on the surface of dust particles, and cloud droplet pH) |
| 7.    Feldspar | 2680 | A fraction of feldspar (K-feldspar) is important for ice nucleation |
| 8.    Quartz | 2670 | It is the most abundant mineral component in silt-sized (diameter:2-63 $\mu m$) minerals. It is important for LW absorption and ice nucleation. |
| 9.    Gypsum | 2308 | It possibly has impact on chemistry, but the impact is likely unimportant given the low abundance globally. |

[1] We use the mean of hematite and goethite densities for hematite, as in Gonçalves Ageitos et al. (2023).

## 231    2.3   AERONET Dust SSA

The AERONET Version 3 Level 2.0 Almucantar inversion retrievals (Giles et al., 2019; Sinyuk et
al., 2020) from 2000 to 2020 are screened for dust events following the methodology in Gonçalves
Ageitos et al. (2023) and Obiso et al. (2023). This screening process aims to select dust-dominated
events and filter out the AERONET scenes contaminated by other absorbing aerosols. The criteria
that are applied to AERONET retrievals to screen dust events are: 1) hourly retrievals from
AERONET are considered to represent dust when the fine volume fraction is small (below 15%),
2) the SSA increases from 440 nm to 675 nm (a feature that distinguishes dust from other species,
see Dubovik et al., 2002), and 3) the mean of the imaginary index at red and near-infrared
wavelengths (675, 870 and 1020 nm) is lower than 0.0042 (as higher values would indicate the
presence of absorbing black and brown carbon, following Schuster et al., 2016). We calculate
AERONET SSA in the visible by averaging AERONET retrieved SSA at two visible wavelengths
(0.44 μm and 0.67 μm) weighted by solar spectrum following Eq. (2).

## 2.4   Laboratory Dust SSA

The lab measured dust SSA at 550 nm is obtained from Di Biagio et al. (2019) (DB-2019 hereafter),
in which dust SSA was directly retrieved from scattering and absorption measurements. We
acknowledge the limits of laboratory measurements, where the dust samples are not aerosols
present in the atmosphere, but instead are reemitted in the lab from soil samples collected from
various source regions. Consequently, the laboratory measurements in DB-2019 do not account
for dust aerosols transported from other regions to the regions of interest. In addition, in contrast
to the modeled dust diameter range of 0.2 $\mu m$ to $20\mu m$, DB-2019 measures dust particles with a
diameter ranging from 0.2 $\mu m$ up to $10\mu m$.

## 2.5   CERES Data

To compare modeled fluxes at TOA with observations, we use the Clouds and the Earth's Radiant
Energy System (CERES) Energy Balanced and Filled (EBAF) Edition-4.2 data (Loeb et al., 2018).
The standard CERES level-3 products provide clear-sky fluxes by averaging all CERES footprints
within a region that are completely free of clouds. Therefore, there are many missing regions in
monthly mean clear-sky TOA flux maps because completely cloud-free conditions are not always
observed at the CERES footprint scale (~20 km at nadir). In contrast to the standard CERES level-
3 products, CERES_EBAF product infers clear-sky fluxes from clear portions of partly cloudy
CERES footprints thereby producing a clear-sky TOA flux climatology free of any missing regions
(details in Loeb et al., 2018). Starting from CERES_EBAF_Ed4.1, the product also provides clear-
sky flux estimates for the total region (i.e., the total CERES footprints) by combining CERES
observations and radiative transfer calculations, which represents clear-sky flux with clouds
removed from the entire atmospheric column of CERES footprints. These clear-sky fluxes for the
total region are defined in a way that is more consistent with how clear-sky fluxes are represented
in climate models (for details see CERES_EBAF_Ed4.1 Data Quality Summary). In this study,
the monthly mean TOA 'Clear-Sky Flux Estimate for Total Region' variables in the
CERES_EBAF_Ed4.2 product, the most recent version of the product, are used to compare with
modeled monthly mean clear-sky flux at TOA. The comparisons allow us to examine the
agreement of modeled clear-sky fluxes from different experiments with observations. The
comparison results will be shown in section 5.1.

## 2.6  CRU TS Data

The CRU TS (Climatic Research Unit gridded Time Series) dataset provides high-resolution (0.5°
latitude × 0.5° longitude) climate dataset over land except Antarctica. The dataset is based on
extensive networks of weather stations going back to 1901(Harris et al., 2020). This dataset has
been widely used in various research areas since its first release in 2000. The mean 2-meter
temperature (TMP) and precipitation rate (PRE) variables from CRU TS v4.07 are used to evaluate
our model simulations. The results will be shown in section 5.2 and section 5.3.

# 3  Experimental Design

Table 2. List of experiments and their description. Experiments are named based on the type of dust used
or the mixing rules for minerals applied in each experiment.

| Experiments | Dust or Minerals | Description | Optics |
|---|---|---|---|
| HD27 | HD27 | Dust refractive index is spatially and temporally uniform. Dust is assumed to contain 2.7% of hematite by volume. Its optical properties are used to represent dust in the standard GFDL AM4.0 model. | Balkanski et al. (2007) |
| HD09 | HD09 | Dust refractive index is spatially and temporally uniform. Dust is assumed to contain 0.9% of hematite by volume. | Balkanski et al. (2007) |
| VOL | VOL-mixing | Soil mineralogy from C1999 is implemented in AM4.0. Hematite (int.) is internally mixed with clay minerals following the volume-weighted mean mixing rule. | Scanza et al. (2015) |
| MG | MG-mixing | Soil mineralogy from C1999 is implemented in AM4.0. Hematite (int.) is internally mixed with clay minerals following the Maxwell Garnett mixing rule. | Scanza et al. (2015) |
| BM | BM-mixing | Soil mineralogy from C1999 is implemented in AM4.0. Hematite (int.) is internally mixed with clay minerals following the Bruggeman mixing rule. | Scanza et al. (2015) |
| BM-RT | BM-mixing | Three experiments are performed step by step to reduce the number of mineral tracers.<br>1) BM-LC experiment: following BM experiment, illite, kaolinite and smectite are lumped together as one tracer 'clay'. | Scanza et al. (2015) |

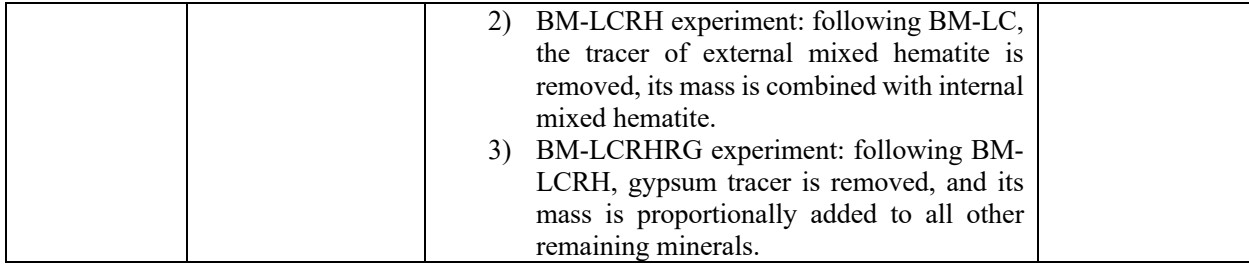

| | | 2) BM-LCRH experiment: following BM-LC, the tracer of external mixed hematite is removed, its mass is combined with internal mixed hematite. <br> 3) BM-LCRHRG experiment: following BM-LCRH, gypsum tracer is removed, and its mass is proportionally added to all other remaining minerals. | |

We conduct a total of six experiments using the GFDL AM4.0 model, with each experiment's description provided in Table 2. Two of these experiments serve as control runs in which dust aerosols are represented with temporally and spatially fixed composition in the model. The first control run, referred to as HD27, represents how dust aerosols are implemented in the standard GFDL AM4.0 model (Zhao et al., 2018a). The second control run is the HD09, in which dust is more scattering than that in the standard AM4.0 model (i.e., HD27) due to its reduced hematite volume fraction from 2.7% to 0.9%.

The other three experiments, namely VOL, MG, and BM, resolve dust mineralogy and activate their interaction with radiation. These three experiments incorporate 45 mineral tracers for nine types of mineral tracers distributed over five size bins as described in Section 2.2. Additionally, we conduct the BM-RT experiments, which consist of three sub-experiments: BM-LC, BM-LCRH, BM-LCRHRG. These experiments aim to explore the potential of reducing mineral tracers, which can improve the model computational efficiency. The results will be discussed in Section 6.

Each of the experiments ran for 19 years, from 2001 to 2019. We consider the 19-year runs of the experiment as a group of simulations, containing 19 members of one-year simulation. The two control runs (i.e., HD27 and HD09), combined with the three mineral-resolved experiments (i.e., VOL, MG, and BM), form a total of six contrasting pairs. In this study, for each contrasting pair, we define the anomaly as the group mean difference (based on 19-year mean) between mineral-resolved experiment and control run. An anomaly is considered statistically significant if the p-value, determined by the student's t-test between the two contrasting groups of simulations, is smaller than 0.05.

# 4 Optical Properties

## 4.1 Dust optical properties

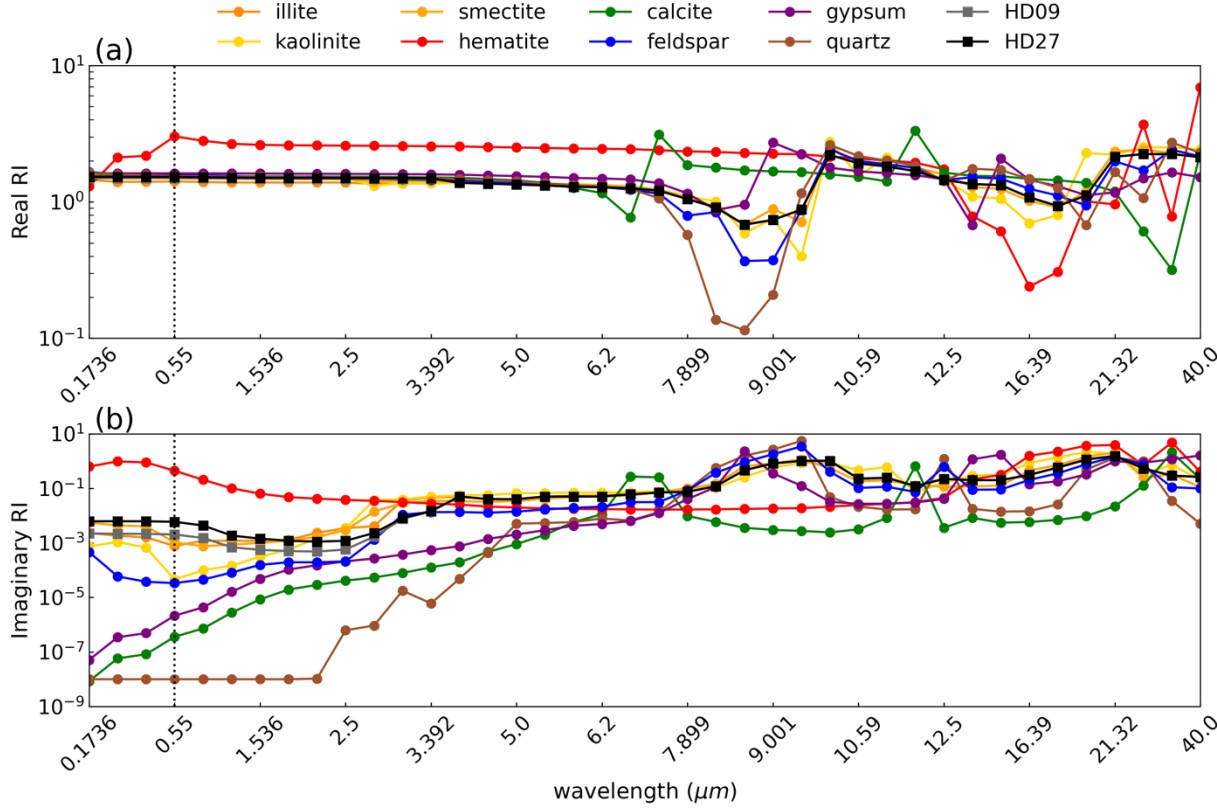

Figure 1. Real part and imaginary part of complex refractive indices (RI) of two homogeneous dust (e.g., HD27 and HD09) and eight minerals (Scanza et al., 2015). The dotted lines represent the real part (a) and imaginary part (b) of dust or minerals for the corresponding wavelength. The black and grey dotted lines are for HD27 and HD09, respectively. The colorful dotted lines are for the eight minerals.

We use the refractive indices (RI) of each mineral from Scanza et al. (2015) and the RI of HD27 and HD09 from Balkanski et al. (2007) to calculate the LUT of optical properties. The spectral RI of each mineral and homogeneous dust (e.g., HD27 and HD09) are shown in Figure 1. The HD09 dust has lower imaginary part of RI at 550 nm than HD27 dust, indicating its lower absorption in the visible band due to a reduced content of hematite.

After calculating LUT of optical properties (details in Section 2.2), we incorporate the interaction of minerals with radiation into GFDL AM4.0. The modeled emission, load, deposition, and lifetime for each mineral are provided in Table S1 in the Supplement. Table 3 provides global total dust emission, load, globally averaged dust aerosol optical depth (DAOD) and SSA for each experiment listed in Table 2 and their comparisons with previous studies. DAOD and SSA from AM4.0 simulations are averaged in the visible band (0.44 - 0.625μm) of GFDL AM4.0. Unless otherwise specified, DAOD and SSA in this study refer to the average in the visible band. Note

that in our calculations, the domain averaged DAOD is always weighted by the area of each grid
cell. The domain averaged SSA is always weighted by the area and DAOD of each grid cell.
Additionally, the spectrally averaged DAOD and SSA are always weighted by the TOA solar
radiation intensity at the corresponding wavelengths, peaking around 0.50 μm, as shown in Eq. (1)
and Eq. (2).

$$\overline{DAOD} = \frac{\int_{\lambda 1}^{\lambda 2} DAOD(\lambda)\, B(\lambda)\, d\lambda}{\int_{\lambda 1}^{\lambda 2} B(\lambda)\, d\lambda} \qquad \text{Eq. (1)}$$

$$\overline{SSA} = \frac{\int_{\lambda 1}^{\lambda 2} SSA(\lambda)\, DAOD(\lambda)\, B(\lambda)\, d\lambda}{\int_{\lambda 1}^{\lambda 2} B(\lambda)\, DAOD(\lambda)\, d\lambda} \qquad \text{Eq. (2)}$$

Where $B(\lambda)$ describes the solar radiation energy intensity, which can be calculated by means of
the Planck's function B(T,λ), using the temperature of the Sun (T = 5800 K).
Table 3. 19-year (2001-2019) averaged global dust emission, load, globally averaged visible band dust
optical depth ($\overline{DAOD}$) and single scattering albedo ($\overline{SSA}$) for each experiment in this study. We use each
grid-cell surface area as a weight for the global DAOD average. We use each grid-cell surface area times
DAOD in each grid-cell as a weight for the global SSA average. In addition, we include the results from
previous studies for the purpose of comparison. Note, the modeled DAOD and SSA in this study are
averaged in the visible band (0.44 - 0.625μm) of GFDL AM4.0, while averaged in the UV-VIS band (0.30
- 0.77μm) of GISS ModelE2.1 in Obiso et al. (2023).

| Experiments | | Emission ($Tg/yr$) | Load ($Tg$) | $\overline{DAOD}$ | $\overline{SSA}$ |
|---|---|---|---|---|---|
| HD27 | | 3354 | 23.6 | 0.022 | 0.86 |
| HD09 | | 3119 | 21.5 | 0.020 | 0.93 |
| VOL | | 3154 | 21.6 | 0.022 | 0.91 |
| MG | | 3083 | 21.1 | 0.021 | 0.93 |
| BM | | 3087 | 21.1 | 0.021 | 0.93 |
| BM-RT | BM-LC | 3097 | 21.1 | 0.021 | 0.930 |
| | BM-LCRH | 3069 | 20.9 | 0.021 | 0.930 |
| | BM-LCRHRG | 3110 | 21.4 | 0.021 | 0.928 |
| AeroCom [1] | | 1600 (1000-3200) | 20 (9-26) | 0.029 (0.021 - 0.035) | - |
| CMIP5[2] | | 2700 (1700-3700) | 17 (14-36) | - | - |

| | | 3472 | 25 | 0.029 | - |
|---|---|---|---|---|---|
| CMIP6[3] | | 3472 | 25 | 0.029 | - |
| DUSTCOMM[4] | | 4700 (3300 - 9000) | 26 (22 - 31) | 0.028 (0.024 - 0.030) | - |
| GISS ModelE2.1 [5] | HOM | 4031 | 31.3 | 0.020 | 0.917 |
| | EXT | 4152 | 32.4 | 0.020 | 0.936 |
| | INT | 4284 | 33.7 | 0.021 | 0.942 |

(1) Results are from AeroCom Phase I, which were taken from Table 3 in Huneeus et al. (2011), and the 1 standard error range was obtained by eliminating the two highest and lowest values.

(2) Results are from Table 3 in Wu et al. (2020)

(3) Results from Zhao et al. (2022)

(4) Results are from Table 3 in Kok et al. (2021)

(5) Results from Obiso et al. (2023).


The lowest SSA of HD27 in Table 3 suggests that HD27 dust, which has been used in the standard
AM4.0 model, is the most absorptive among all experiments. The HD09 dust is much less
absorptive, attributed to its smaller hematite content, as indicated by the lower imaginary part of
RI in the visible range (Figure 1). For the three mineral-resolved experiments, the lower global
mean SSA ($\overline{SSA}$) in VOL suggests that VOL-mixing dust is more absorptive than MG-mixing
and BM-mixing dust. This finding is consistent with previous studies that have suggested that
VOL-mixing method, when applied to minerals to compute the bulk aerosol optical properties,
may artificially enhance absorption relative to scattering and lead to a lower SSA for bulk dust
aerosol (Zhang et al., 2015). We can see that the global mean SSA ($\overline{SSA}$) of HD09 dust is
comparable to the values obtained in cases where minerals are resolved (e.g., MG and BM). This
implies that, from a global perspective, HD09 dust is as absorptive as mineral-resolved dust (e.g.,
MG and BM).
In addition to the globally averaged dust properties listed in Table 3, we illustrate the distribution
of global dust mass across 5 size bins (Figure S3 in the Supplement) and the global distribution of
DAOD (Figure 2) for the three experiments: before (e.g., HM27 and HD09) and after (e.g., BM)
resolving mineralogy. The global dust mass distribution across the 5 size bins remains largely
unchanged across experiments. Besides the subtle difference (~10%) in global mean DAOD across
the three experiments as listed in Table 3, the global distribution of DAOD responds differently in
HD09 and BM. Compared to HD27, reducing hematite content to HD09 generally decreases
DAOD, except over the Sahel region. In contrast, resolving mineralogy as in BM decreases DAOD
over the Sahara region while increasing DAOD over the Sahel and Asia regions. The reduction in
DAOD over the Sahara region further contributes to the decrease in dust absorption over the region,
primarily attributed to the change in dust optical properties, such as the enhancement in dust SSA.
The indistinct variation in DAOD across different experiments results from the feedback of dust
interactions with radiation (Miller et al., 2004; Pérez et al., 2006; Miller et al., 2014), which is
influenced by the distinct scattering properties of dust aerosols in each experiment as shown in
Table 3.

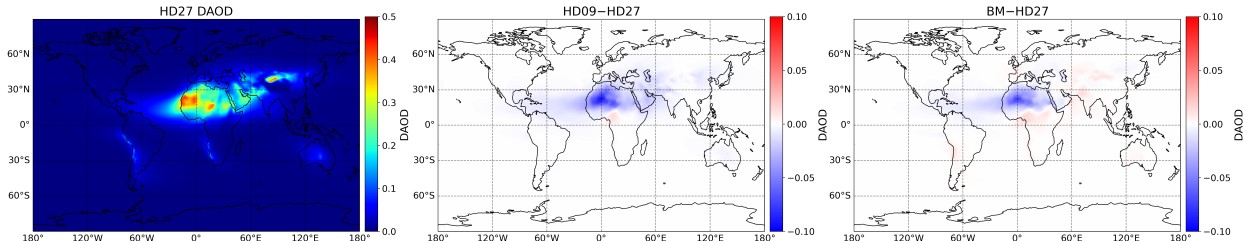


Figure 2. The global distribution of dust optical depth (DAOD) for HD27, and the difference in DAOD
between HD09 and BM compared to HD27. The global mean DAOD values ($\overline{DAOD}$) of each experiment
are shown in Table 3.

## 4.2 Comparison of dust optical properties with observations

Iron oxides content of dust determines shortwave radiation absorption by dust: the higher amount
of iron oxides, the lower the SSA. Following our calculations of dust optical properties in Section
4.1, we compared GFDL AM4.0 modeled dust SSA (averaged in the visible band 0.44-0.625 μm)
against AERONET SSA retrievals (averaged at two visible wavelength: 0.44 μm and 0.67 μm) in
Section 4.2.1 and laboratory measurements of SSA (at 0.55 μm) in Section 4.2.2. The modeled
dust SSA is evaluated against observation-based results utilizing the following evaluation metrics:
the mean SSA (mSSA) is calculated based on SSA for all locations displayed in Figure 3, the
standard deviation (σ), derived from SSA for all locations displayed in Figure 3, is used as an
indicator of dust SSA spatial variation; the normalized mean bias ($nMB$) and normalized root
mean square error ($nRMSE$) are utilized to assess the mean bias and root mean square error,
respectively, of modeled SSA in comparison to observed SSA. Definitions of $nMB$ and $nRMSE$
are provided in Section S3 in the Supplement.

## 4.2.1   Comparison with AERONET retrievals

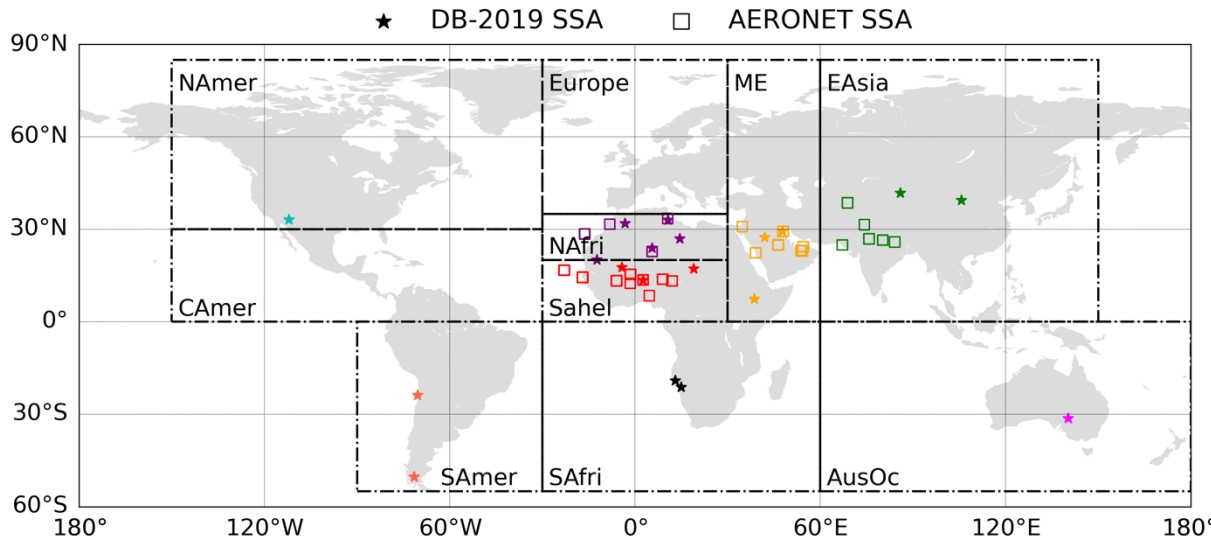

Figure 3. Dust sample locations in Di Biagio et al. (2019) and AERONET stations selected by filtering dust events. AERONET stations for SSA retrievals are in North Africa (NAfri), Sahel, the Middle East (ME) and East Asia (EAsia). Lab measurements by Di Biagio et al. (2017, 2019) expand dust sampling to include soils from North America (NAmer), South America (SAmer), South Africa (SAfri), Australia (AusOc).

Figure 3 displays the AERONET stations selected by filtering dust events. The global distribution of modeled dust SSA and AERONET retrieved SSA over the selected AERONET sites are shown in Figure 4. There is a significant decrease in dust absorption from HD27 to HD09 globally due to the reduction in hematite content.  HD09 and BM exhibit similar dust absorption on a global scale (e.g., the same global mean $\overline{SSA}$), but the regional differences are evident. For instance, compared to HD09, resolving mineralogy (e.g., BM) decreases dust absorption over Iceland and Taklamakan regions, while enhances dust absorption over Southern Hemisphere, particularly over Australia. Additionally, there is a shift in dust absorption from the Sahara to the Sahel region after resolving mineralogy.

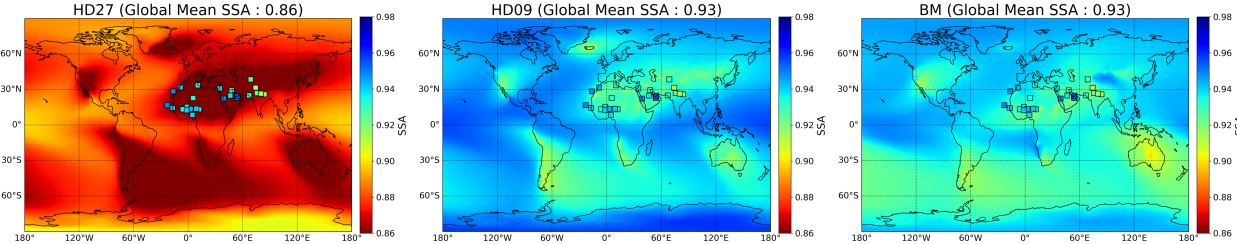

Figure 4. The 19-year (2001-2019) annual mean dust SSA simulated by AM4.0 across the three experiments. The squares represent 21-year (2000-2020) annual mean AERONET retrieved dust SSA over the selected AERONET sites.

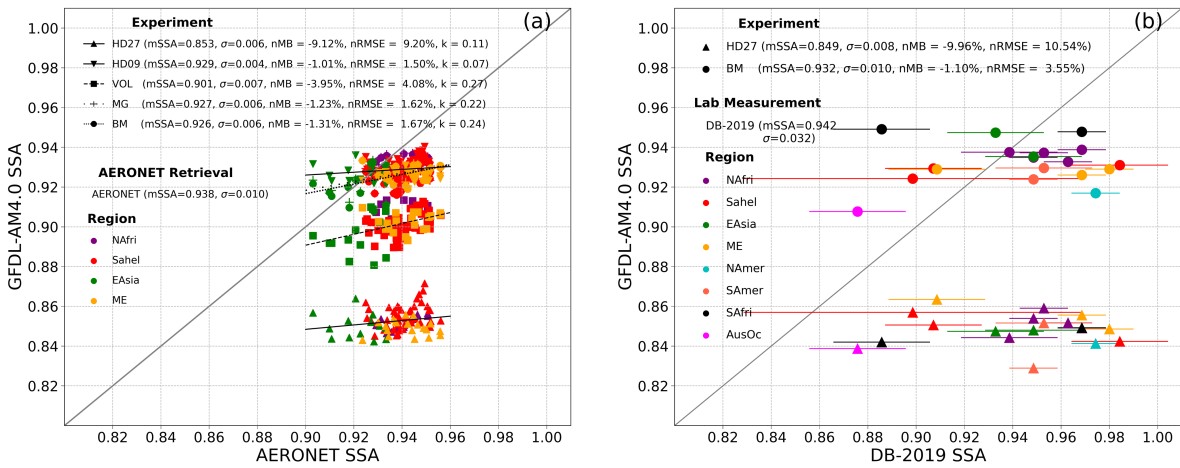


Figure 5. GFDL AM4.0 modeled 19-year (2001-2019) averaged monthly dust SSA (average in 0.44-0.625
μm) versus (a) AERONET 21-year (2000-2020) averaged monthly SSA retrievals (average at two visible
wavelengths: 0.44 μm and 0.67 μm) and (b) laboratory SSA measurements (at 0.55 μm) of dust particles
with diameter ranging from 0.2 $\mu m$ up to $10\mu m$ obtained by Di Biagio et al. (2019) (DB-2019). The lab
measurements were carried out in March 2015, horizontal error bars represent measurement uncertainties.
Markers represent different experiments, and colors represent different regions. mSSA in the legend
represents the mean SSA averaged over all locations indicated in Figure 2 (squares for AERONET, stars
for DB-2019). The standard deviation (σ), normalized mean bias (nMB), normalized root mean square error
(nRMSE), and the slope of linear regression (*k*) are also indicated in the legend.
Figure 5a shows GFDL AM4.0 modeled 19-year (2001-2019) averaged dust SSA (average in 0.44-
0.625 μm) versus AERONET SSA (average at 0.44 μm and 0.67 μm) retrievals. Compared to
AERONET SSA retrievals, both HD27 and VOL overestimate dust absorption, as indicated by
their relatively low SSA (Figure 5a). HD27 dust is the most absorptive, indicating that the standard
AM4.0 dust is overly absorptive. Dust SSA in MG and BM are quite similar (i.e., mSSA: 0.926
versus 0.927) and show a better agreement with AERONET measurements ($nMB \approx -1.3\%$ and
$nRMSE \approx 1.6\%$), and they exhibit stronger scattering (i.e., higher SSA) than HD27 and VOL.
HD09 is almost as scattering as MG and BM, as indicated by the mSSA of 0.929 versus 0.926 and
0.927 in Figure 5a, which is consistent with the global mean SSA ($\overline{SSA}$) results shown in Table 3.
Overall, both the fixed mineralogy dust HD09 and mineral-resolved MG and BM dust agree well
with AERONET SSA retrievals, while HD27 and VOL are too absorptive. Therefore, we
recommend using mixing ratios of MG or BM in calculating optical properties of internal mixture
of hematite and clay minerals. Unless otherwise specified, in the following part of the paper, we
will refer to mineral-resolved experiments as MG or BM.
We further assess the SSA spatial variation (indicated by σ) for each experiment from AM4.0 by
comparing it to observation-based results. SSA is generally determined by dust mineralogy, size
as well as shape. Various dust mineralogy leads to distinct dust SSA due to the different absorption
properties of minerals (Figure 1). Coarser dust generally tends to be more absorptive (i.e., has
lower SSA) than finer dust when other factors are the same (Ryder et al., 2018). Spherical dust
assumption tends to underestimate dust SSA (Huang et al., 2023). Given the uncertainty in dust
shape, we assume dust particles to be spherical in this study, aligning with other model studies
(e.g. Gliß et al., 2021). Consequently, in the mineral-resolved experiments of this study, namely
VOL, MG, and BM, dust mineralogy and dust size are the two factors affecting the SSA spatial
variation.
Conversely, in homogeneous dust experiments, specifically HD27 and HD09, SSA variation is
solely attributed to variation in dust size, as dust mineralogy remains uniform globally.
Interestingly, HD09 demonstrates smaller spatial variation (i.e., lower σ) in SSA compared to
HD27 (Figure 5a). To investigate the impact of dust size on SSA for different hematite content
(e.g., HD27 and HD09), we perform a simple test in Section S4 of the Supplement. Supplementary
Figure S4 and S5 illustrate that the variation of SSA due to the dust particle size is more
pronounced with increasing absorption, i.e., from HD09 to HD27. This suggests that enhancement
in dust scattering relative to dust absorption (i.e., an increase in SSA) mitigates the sensitivity of
SSA to dust size.
The conclusions above provide an understanding of the SSA spatial variation (indicated by σ)
before (i.e., HD27 and HD09) and after (i.e., VOL, MG, and BM) implementing dust mineralogy.
The same σ (0.06) between HD27 and BM can be explained as follows: Because mineral-resolved
BM-mixing dust is overall more scattering than HD27 dust, resulting in a reduced sensitivity of
SSA to size, therefore, the σ of SSA caused by dust size is reduced in BM relative to HD27.
However, the incorporation of dust mineralogy in BM leads to an increase of σ. These contrasting
effects compensate for each other, resulting in the same σ. In contrast, HD09 is overall as
scattering as BM, as shown in Table 3and Figure 5a, suggesting a similar sensitivity of SSA to
size. Therefore, the incorporation of dust mineralogy in BM results in a higher σ compared to
HD09. Overall, while the enhancement in σ can be offset by the reduction in σ due to the reduced
sensitivity to dust size, resolving dust mineralogy increases $\sigma$ on its own, consequently enhancing
the spatial variation in dust SSA.
Worth to mention, AERONET dust is quite scattering as shown in Figure 5a, therefore its SSA is
less sensitive to dust size. The high $\sigma$ (0.010) of AERONET SSA can be mainly due to spatial
variations in dust mineral composition. Reducing dust hematite content (HD09) leads to a better
agreement in mean SSA with AERONET (i.e., more scattering dust) but results in very low $\sigma$
(0.004), while implementing dust mineralogy (e.g., MG and BM) retains the agreement with
AERONET in mean SSA and, at the same time, increases $\sigma$ (0.006).
Besides the standard deviation ($\sigma$), the slopes ($k$) obtained from the statistics of modeled SSA
versus AERONET SSA can also indicate the regional contrast of SSA. The regional contrast of
AERONET SSA is well captured by the model when $k$ is one, underestimated when $k$ is lower than
one, and overestimated when k is higher than one. As such, the slopes in Figure 5a show that the
contrast in SSA from different regions (e.g., North Africa vs. East Asia) observed by AERONET
is better captured by mineral-resolved experiments (e.g., VOL, MG and BM with $k$ ranging from
0.22 to 0.27) than homogeneous dust experiments (e.g., HD27 and HD09 with $k$ ranging from 0.07
to 0.11). However, the modeled regional contrast of SSA in mineral-resolved experiments remains
overly underestimated (i.e., $k$ is much lower than one).  The significant underestimation of regional
SSA contrast ($k$) and spatial variation ($\sigma$) in AM4.0, even after accounting for mineralogy, implies
that something important is still missing in models. For instance: 1) the observed regional contrast
in iron oxides content may be higher than that in the soil map used in this study, and 2) the model
may have underestimated regional contrasts in the dust aerosol size distribution and thus their
contribution to SSA, and 3) spatial variation of dust shape, which is not taken into account in the
model.
4.2.2   Comparison with laboratory measurements
We further compare the GFDL AM4.0 modeled dust SSA (average in 0.44-0.625 µm) with DB-
2019 laboratory measurements of SSA at 0.55µm (Figure 5b). Figure 3 shows the locations where
dust samples were collected for the lab measurements. Considering that MG and BM are very
similar in terms of dust absorption and agree the best with AERONET SSA, we select BM as a
representative to compare with DB-2019 SSA. Moreover, to evaluate how dust absorption is
represented in the standard AM4.0 relative to lab measurements, we also show the comparison of
SSA between HD27 used in the standard AM4.0 and DB-2019 (Figure 5b). Consistent with the
comparison with AERONET, the comparison with lab measurements suggests that dust
representation in the standard AM4.0 (i.e., HD27) is excessively absorptive. The smaller nMB and
nRMSE values in BM suggests that SSA of BM agrees better with lab measurements.
Moreover, regarding spatial variation ($\sigma$), resolving dust mineralogy in BM increases $\sigma$ from 0.008
for HD27 to 0.010 for BM, even though it is still lower than the $\sigma$ (0.032) in DB-2019 lab
measurements. Note that the variation for HD27 results from the high sensitivity of SSA to dust
size due to its higher absorption (as discussed in Supplementary Section S4). The inability to
reproduce spatial variation observed in the lab measurements is likely attributed to two aspects.
The first limitation is the fact that samples of DB-2019 are from soils rather than aeolian dust.
Aeolian dust is expected to exhibit greater uniformity in mineralogy than soils because of the
atmospheric mixing of dust emitted from various soil sources.  The second one is associated with
the under-representation of regional contrast in iron oxides content in our model. Observations
from the EMIT are therefore essential to constrain soil mineralogy in climate models.

## 5    Impacts of dust mineralogy on climate

Resolving dust mineralogy in climate models affects dust optical properties (as discussed in
Section 4) and their spatial and temporal variability, thereby affecting their interactions with
shortwave (SW) and longwave (LW) radiation. The variability in dust radiative interactions further
induces the fast response of land surface temperature, circulation, and precipitation. To investigate
the impacts of resolving dust mineralogy on climate, we need to compare modeled results in
mineral-resolved experiments to the baseline homogeneous dust (i.e., non-resolved mineralogy)
control run. As a result, the significance of the impacts depends on the selection of the baseline
homogeneous dust experiment. In this section, we investigate the impacts of resolving dust
mineralogy compared to two baseline homogeneous dust experiments. One baseline experiment is
the homogeneous dust used in the standard GFDL AM4.0, in which dust mineralogy is assumed
to be temporally and spatially uniform (i.e., non-resolved), with a volume fraction of 2.7%
hematite (HD27). The impacts of resolving dust mineralogy, compared to the baseline HD27, can
be attributed to two factors: 1) the reduction in dust absorption after resolving dust mineralogy in
comparison to HD27 as discussed in Section 4, and 2) spatial and temporal variations in dust
scattering properties induced by inhomogeneity in dust mineralogy.  The other baseline experiment
is the homogeneous dust, with a volume fraction of 0.9% hematite (HD09). Given the comparable
global mean dust scattering properties (e.g., $\overline{SSA}$) between the control run HD09 and mineral-
resolved experiments (e.g., MG and BM), the impacts of resolving dust mineralogy, compared to
baseline HD09, is solely attributed to the inhomogeneity in dust scattering properties induced by
resolving dust mineralogy.

## 520    5.1   Impacts on Clear-sky Radiative Fluxes

We start our analysis by examining the impacts of resolving dust mineralogy on clear-sky radiative
fluxes. By 'clear-sky', we mean that our results do not consider the radiative effects of clouds. We
use anomalies ($\Delta F$) to evaluate the impacts of resolving mineralogy on clear-sky radiative fluxes,
which is defined as, $\Delta F = F_1^\uparrow - F_2^\uparrow$ , where $F_1^\uparrow$ is the 19-year mean clear-sky upward radiative
fluxes with resolved mineralogy, $F_2^\uparrow$ is the 19-year mean clear-sky upward radiative fluxes for
homogeneous dust. Section S5 in the Supplement provides the clear-sky radiative fluxes anomalies
at TOA and surface (SFC) induced by resolving dust mineralogy over the global scale, we see
much more significant anomalies over the North Africa than other regions, which makes senses
because that dust aerosol is the most dominant aerosol species in this area. The changes in dust
aerosol optical properties have a greater potential to lead to significant impacts on radiation and
climate over the region than in the others. Therefore, this section focuses on the North Africa
region, where the Sahara Desert, the largest dust source in the world, is located. The Sahara (20°N-
30°N, 10°W-35°E) and the Sahel (10°N-20°N, 10°W-35°E) regions are studied separately. We
specifically analyze the results for the June-July-August (JJA) season when dust loading is at its
highest and the West African Monsoon is the strongest.

### 536    5.1.1   Impacts on Clear-sky Radiative Fluxes relative to HD27

Before comparing mineral-resolved experiments (e.g., BM) with HD27 control run, the dust has
been used in the standard GFDL AM4.0, to understand their impacts on clear-sky radiative fluxes
relative to HD27, it's worth recapping that the effects of resolving mineralogy relative to HD27
can be attributed to two factors: the reduction in dust absorption and the variation in dust scattering
properties induced by the mineralogical inhomogeneity.

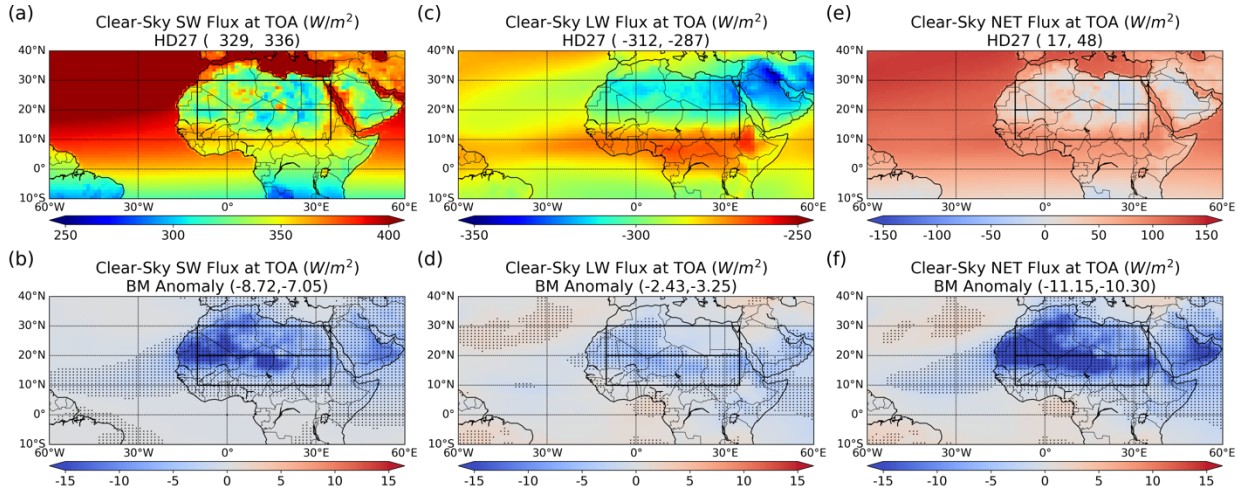

Figure 6. Seasonal mean JJA climatology (2001-2019) clear-sky SW (1st column), LW (2nd column) and Net (3rd column) radiative flux at TOA for the HD27 control run (1st row) and their anomalies resulting from resolving dust mineralogy in Bruggeman-mixing experiment (2nd row). Downward direction is defined as positive. The dotted area denotes anomalies that are statistically significant. The two values in parentheses within the title of each figure are domain average for the Sahara and Sahel regions.

The first row in Figure 6 illustrates the modeled clear-sky shortwave (SW), longwave (LW) and net (NET: the combination of SW and LW) radiative flux at TOA from the HD27 control run. Relative to HD27, mineral-resolved dust (e.g., BM-mixing dust) generally reflects more SW radiation back to space and induces negative SW flux anomalies at TOA (Figure 6a, b; Positive: downward). Relative to the HD27 control run, the LW flux anomaly at TOA resulting from resolving mineralogy is less substantial compared to SW flux anomaly (Figure 6c, d). After combining both SW and LW, resolving mineralogy turns out to induce substantial decrease in NET flux at TOA, with a more than 50% negative anomaly over the Sahara and around a 20% negative anomaly over the Sahel (see values in parentheses in Figure 6e, f). Therefore, less NET radiation reaches the Earth at TOA in the mineral-resolved dust cases due to their lower absorptivity.

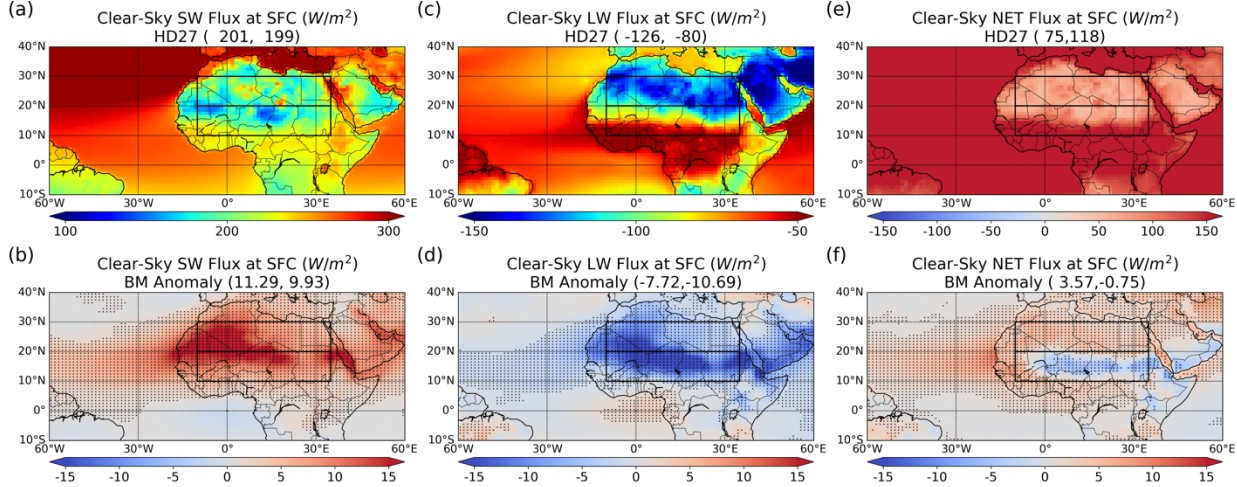


Figure 7. As in Figure 6, but for the surface.
At the surface (SFC) in Figure 7, the enhanced scattering of mineral-resolved dust scatters more
SW radiation toward Earth's surface, leading to a positive SW flux anomaly at SFC (Positive:
downward). In the LW, the cooling of the mineral-resolved dust layer, due to its low absorption,
results in less LW radiation being emitted toward Earth's surface. This reduction in the downward
LW emission outweighs the change in the upward LW emission from the Earth's surface, thereby
causing a negative LW flux anomaly at SFC. The positive anomalies in SW radiation are
approximately canceled out by the negative anomalies in LW radiation (Figure 7). As a result, a
similar amount of radiation reaches the Earth's surface in both HD27 and mineral-resolved cases.
Despite less NET radiation entering the Earth at TOA in mineral-resolved cases, the similar
amount of NET radiation reaching the Earth's surface indicates that less NET radiation is absorbed
in the atmosphere in mineral-resolved cases.

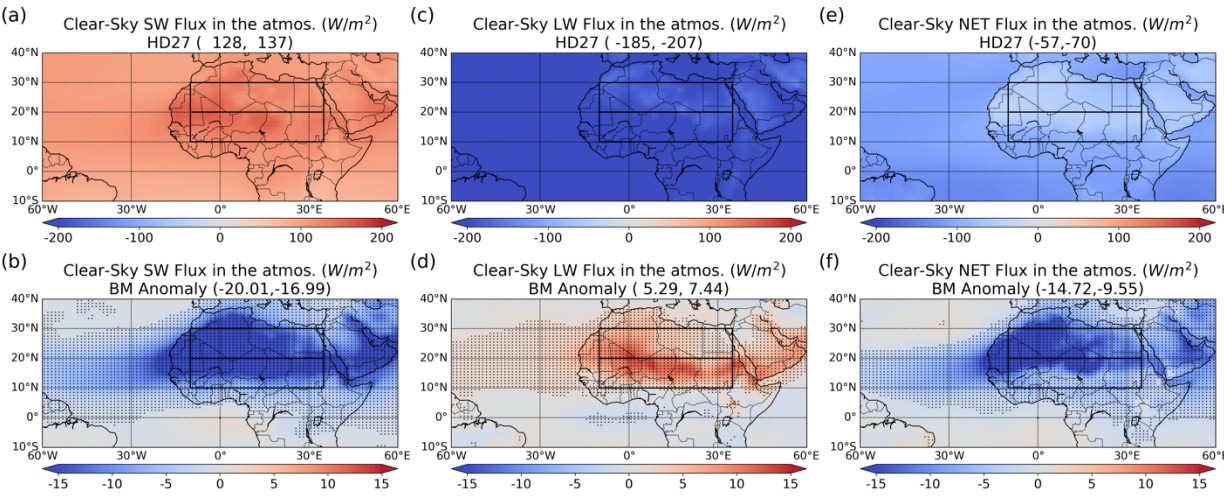


Figure 8. As in the Figure 6, but for the radiative flux absorbed in the atmosphere.
In Figure 8, the negative SW flux anomalies are partially offset by positive LW flux anomalies,
resulting in negative NET flux anomalies in the atmosphere. These anomalies amount to
approximately a 25% reduction over the Sahara and 10% reduction over the Sahel (see values in
parentheses in Figure 8b, d, f).
5.1.2   Impacts on Clear-sky Radiative Fluxes relative to HD09
Before comparing mineral-resolved experiments with HD09 control run, where the homogeneous
dust is as absorptive as mineral-resolved dust (e.g., MG and BM) from a global perspective, to
understand their impacts on clear-sky radiative fluxes relative to HD09, it's worth recapping that
the effects of resolving mineralogy relative to HD09 are primarily attributed to the variation in
dust scattering properties induced by the mineralogical inhomogeneity.

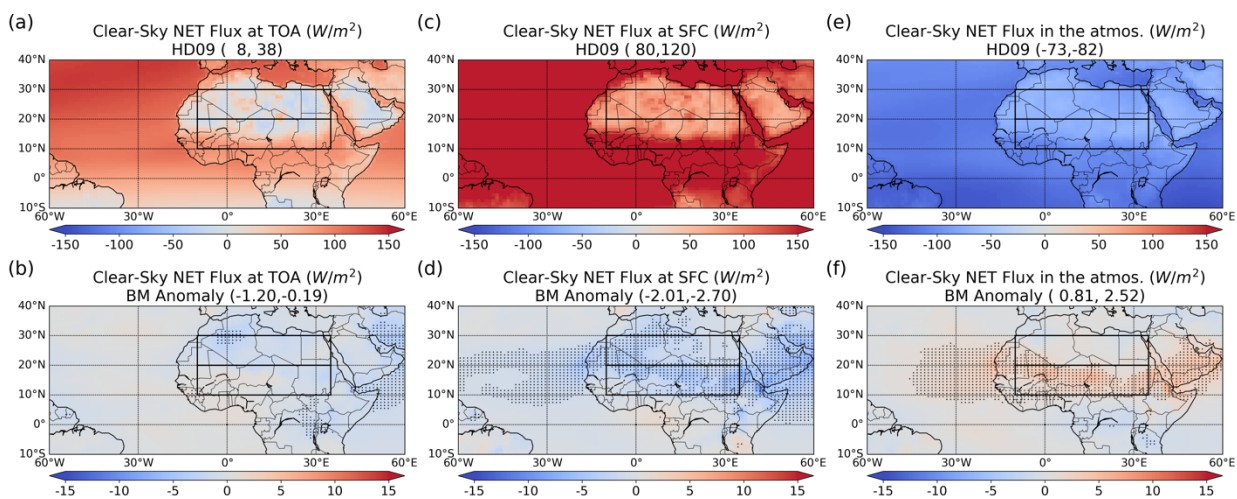


Figure 9. As in the Figure 6, but for HD09 control run. In addition, SW and LW flux anomalies are not
shown here. Clear-sky net flux at TOA (1$^{st}$ column), at surface (2$^{nd}$ column), and in the atmosphere (3$^{rd}$
column) are shown in this figure.
**Error! Reference source not found.** Figure 9 shows the clear-sky fluxes anomalies with respect t
o HD09 over North Africa, the anomalies over the global scale are shown in Figure S7, S9, S11
and S13 in the Supplement.
In contrast to the anomalies with respect to HD27 control run, resolving dust mineralogy does not
cause substantial anomalies ($< 5\%$) in clear-sky fluxes with respect to HD09 control run. This can
be attributed to their similarity in dust scattering properties from a global mean perspective,
particularly SSA as shown in Figure 5. The comparable effects of HD09 and mineral-resolved dust
on radiation suggest that resolving dust mineralogy does not have significant impacts on clear-sky
fluxes when homogeneous dust is as scattering as mineral-resolved dust aerosols on a global scale.
However, the equivalence between HD09 and mineral-resolved dust in terms of their interactions
with radiation may be related to the three limitations in the current model simulations: 1) Soil
mineralogy: The limited soil mineralogy database fails to adequately capture the regional variation
of iron content (or SSA) within the region; 2) Dust emission based on Ginoux et al. (2001) uses a
continuous function of topography, which does not take into account geomorphological
characteristics of the surface to differentiate soil properties of dust sources as done by others
(Zender et al., 2003; Bullard et al., 2011); 3) Dust transport: Excessive numerical diffusion may
occur when solving advection equation (Ginoux, 2003). Given all those limitations of our model
simulations, this finding may differ with improved representation of dust sources and transport.
Such improvement may come from spaceborne soil mineralogy dataset (e.g., EMIT) that may
capture accurately the regional contrasts in iron oxides content.
### 5.1.3   Compare Clear-sky Radiative Fluxes with CERES Observations
Furthermore, we conduct a comparison of modeled SW upward, LW upward, NET downward flux
at TOA with observation-based results from CERES_EBAF_Ed4.2 product (see Table 4). The
difference between modeled flux and CERES observations are listed in parentheses within the title
of each figure in Table 4. Compared to HD27, the more scattering HD09 and mineral-resolved BM
achieve much better agreement with CERES observations in clear-sky flux (i.e., SWup, LWup and
NETdn) at TOA. This is evident in the smaller values of HD09 – CERES (e.g., NETdn: 1.6 for the
Sahara and 2.4 for the Sahel) and BM – CERES (e.g., NETdn: 0.4 for the Sahara and 2.1 for the
Sahel) compared to HD07 – CERES (e.g., NETdn: 11.3 for the Sahara and 12.4 for the Sahel), as
shown by the values in parentheses in Table 4. Between HD09 and BM, BM tends to agree slightly
better with CERES.
Table 4. Comparison of modeled clear-sky SW upward (SWup, 1$^{st}$ row), LW upward (LWup, 2$^{nd}$ row) and
NET downward (NETdn, 3$^{rd}$ row) fluxes at TOA with CERES observation-based results over 2001-2019
JJA. The 1$^{st}$ column shows the clear-sky flux estimates at TOA from CERES_EBAF_Ed4.2 product, which
represents clear-sky flux with clouds removed from the atmospheric column. The following columns show
the difference of modeled clear-sky flux at TOA in HD27 (2$^{nd}$ column), HD09 (3$^{rd}$ column) and BM (4$^{th}$
column) experiments from CERES observations. The two values in parentheses represent domain average
for the Sahara and Sahel regions as indicated in figures in Section 5.1.2. Specifically, the first column
(CERES) is domain averaged flux, while the second (HD27 – CERES), third (HD09 – CERES), and fourth
(BM – CERES) columns are domain averaged flux differences between model and CERES observation-
based results.

|  | CERES | HD27 – CERES | HD09 – CERES | BM – CERES |
|---|---|---|---|---|
| Clr SWup flux at TOA (W/m$^2$) | (135, 113) | (–9.3, –9.4) | (–1.1, –2.2) | (–0.6, –2.3) |
| Clr LWup flux at TOA (W/m$^2$) | (314, 291) | (–1.4, –3.1) | (–0.6, –0.5) | (0.2, –0.1) |
| Clr NETdn flux at TOA (W/m$^2$) | (6, 36) | (11.3, 12.4) | (1.6, 2.4) | (0.4, 2.1) |


## 5.2   Impacts on land temperature

Here we explore the impacts on the temperature vertical profile and near-land surface temperature
relative to HD27 and HD09, respectively. Compared to the HD27 control run, lower absorption of
radiation in the atmosphere by mineral-resolved dust aerosols results in statistically significant
negative temperature anomalies in the atmosphere ranging from 800 mb up to 500 mb where dust
aerosols are mainly located (Figure 10). In contrast, there is no statistically significant temperature
anomaly for mineral-resolved dust cases compared to HD09, as illustrated by the red curves in
Figure 10 This finding aligns with the insubstantial anomalies (<5%) in clear-sky NET radiative
fluxes discussed in Section 5.1.2. In the subsequent part of the section, we will delve into
comparing mineral-resolved experiment (using BM as an example) with the HD27 control run.
This comparison will help us further understand the impact of dust aerosols with distinct
absorption on land temperature.

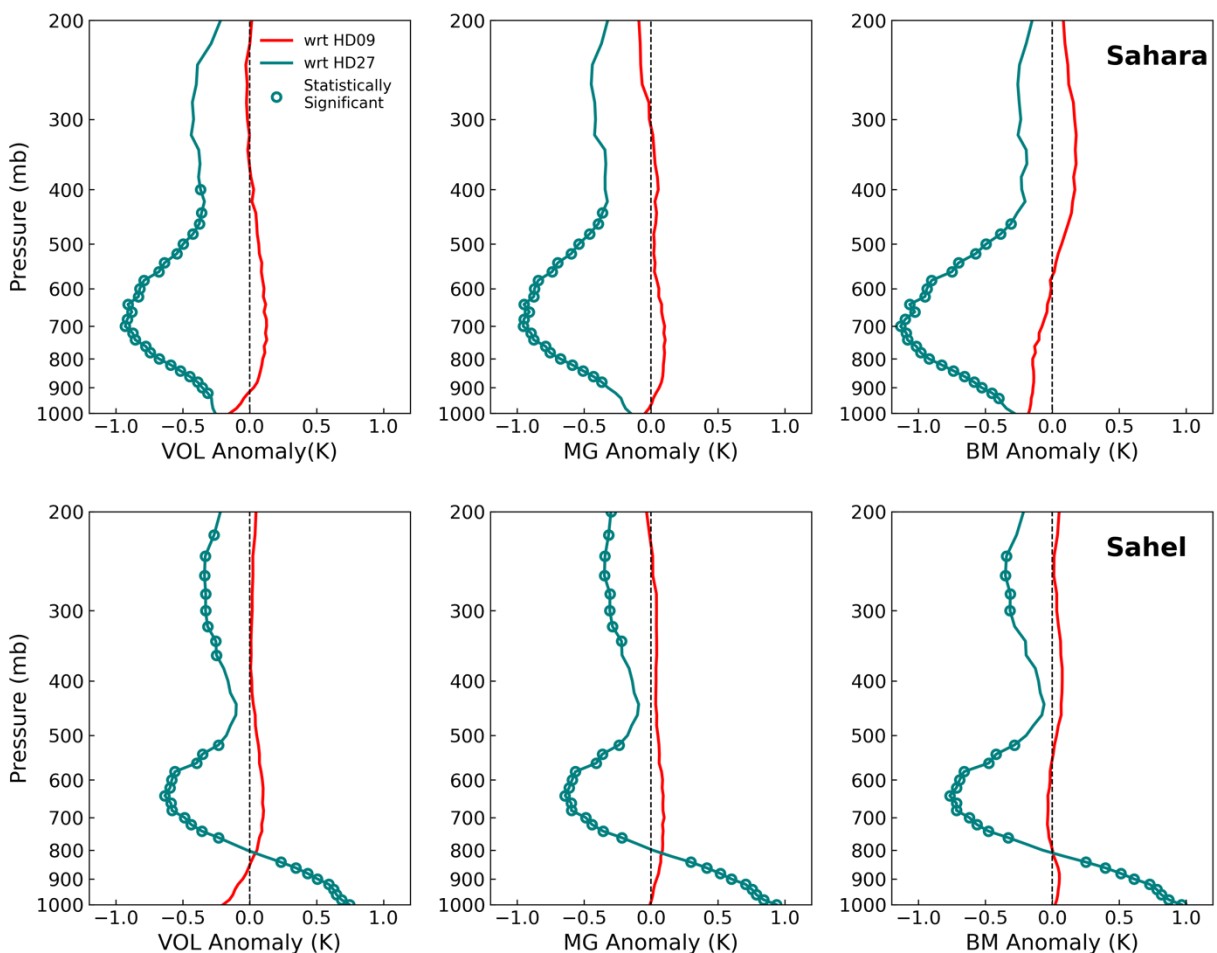

Figure 10. Vertical profile of temperature anomaly induced by resolving dust mineralogy for the Sahara (1st row) and the Sahel (2nd row) regions in the three mineral-resolved experiments (i.e., VOL, MG, BM). Green lines represent temperature anomalies with respect to HD27 control run. Red lines are temperature anomalies with respect to HD09 control run. The circles represent statistically significant temperature anomaly.

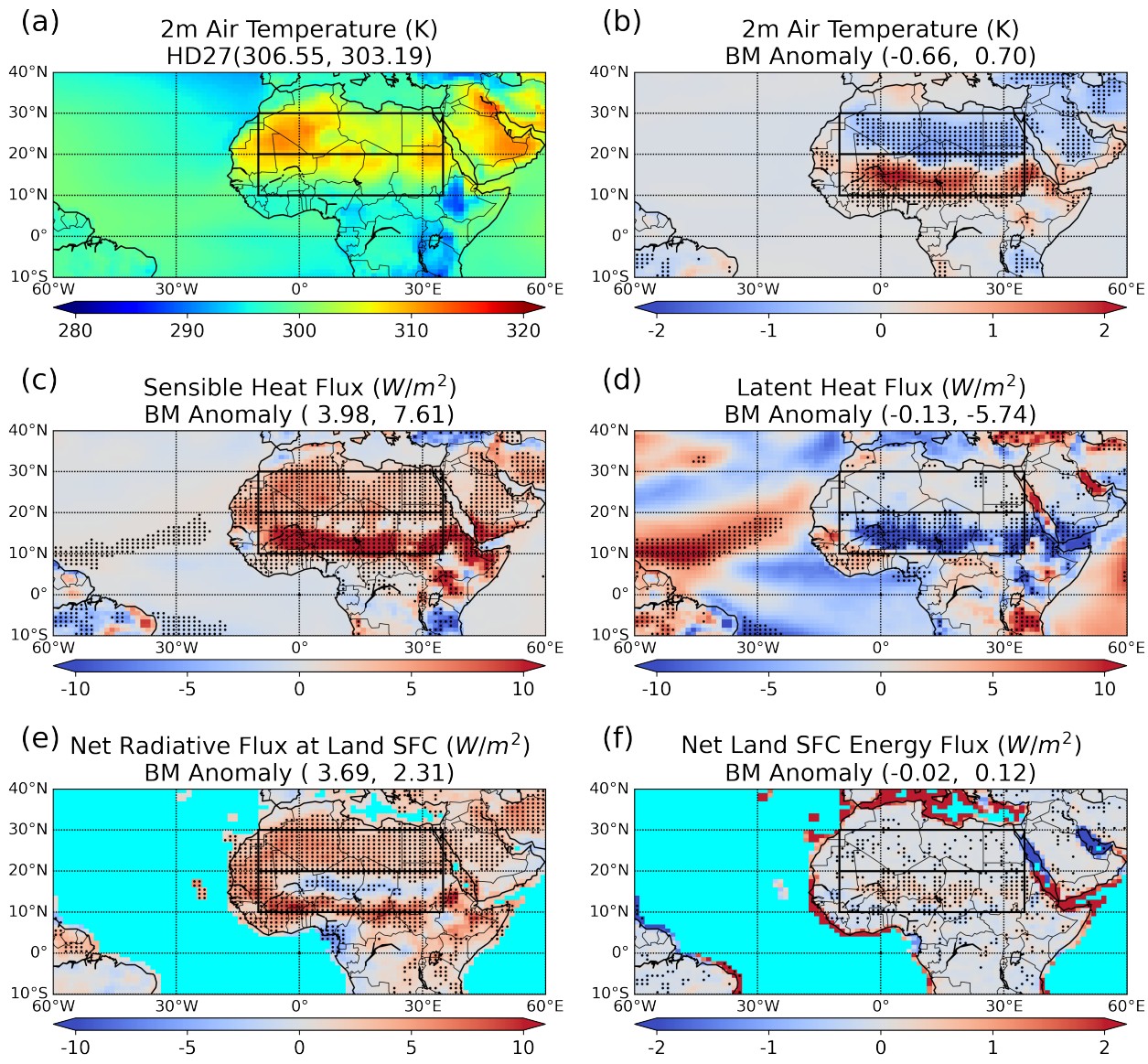

Figure 11. Air temperature at 2-meter from HD27 control run (a), anomaly (b) induced by implementing Bruggeman-mixing minerals in BM experiment, surface sensible heat flux (c), latent heat flux (d), net radiative flux (e), net energy flux (f) anomalies between BM and HD27; Upward flux is positive in (c) and (d), while downward flux is positive in (e) and (f). Net energy flux (f) is the subtraction of (c), (d), and downward ground flux from (e). Note that ground flux is not shown in the figure considering its relatively small magnitude, but it is included in the land surface net energy flux calculations in subplot (f). The dotted area denotes anomalies that are statistically significant. The two values in parentheses within the title of each figure are domain average for the Sahara and Sahel regions.

Figure 11a shows air temperature at 2-meter from HD27 control run over the Northern Africa. Near the land surface, more scattering mineral-resolved dust induces a temperature decrease (i.e., negative temperature anomaly –0.66 K) over the Sahara and a temperature increase (i.e., positive

temperature anomaly 0.70 K) over the Sahel as shown Figure 11b. To understand this phenomenon,
we further analyze the surface energy budget in Figure 11c-f.
Radiative flux perturbation over land is quickly equivalented by balancing surface radiative fluxes
with sensible heat flux, latent heat flux and ground flux (i.e., downward heat flux into the ground),
which results in nearly zero net energy flux at land surface as shown in Figure 11f. Precisely, the
radiative flux anomaly comprises two contributions: one is the instantaneous radiative forcing (IRF)
caused by the change in dust mineralogy in the atmosphere, and the other one is the associated
radiative feedbacks. For simplicity, we will not partition the radiative flux anomaly in our
discussion here. Over the Sahel region, the positive net radiative flux anomaly at land surface is
balanced out by the increased sensible heat flux and the decreased latent heat flux as well as ground
flux. Note that the ground flux is generally small in magnitude and not shown in Figure 11, but we
include it in calculating net surface energy flux in Figure 11f. The decrease of latent heat flux over
the Sahel in BM case (Figure 11d) is due to the depletion of soil moisture (and therefore
evaporation) in the region as shown in Figure 12. The depletion of soil moisture is caused by the
decrease in moisture carried by onshore winds over the Sahel and the decrease in precipitation
over the same region, as will be discussed in section 5.3. Therefore, a large enhancement of
sensible heat flux ($\sim$7.6 $W/m^2$) is needed (Figure 11c) not only to compensate for the depletion
in latent heat flux ($\sim$5.7 $W/m^2$ in d), but also to balance out the increased net radiative flux ($\sim$2.3
$W/m^2$ in Figure 11e). As a result, higher land surface temperature with anomaly around 0.7 K is
needed in the region to achieve the required sensible heat flux enhancement.

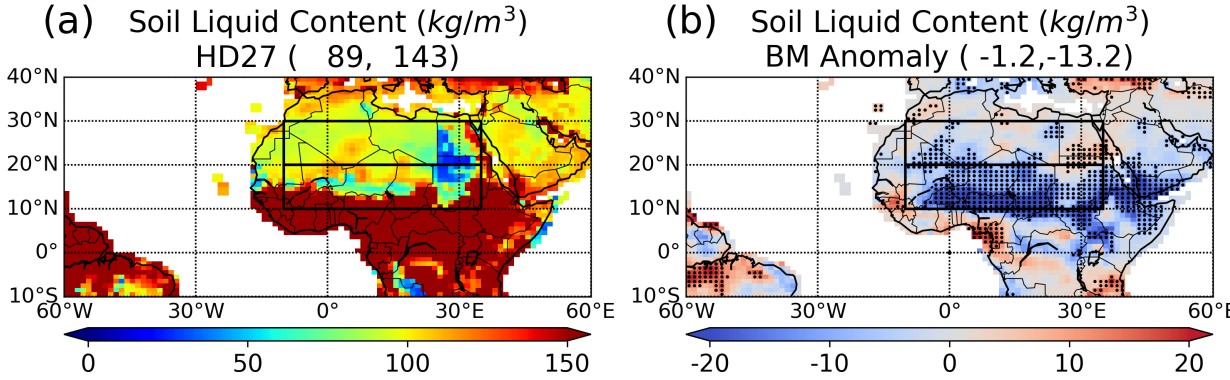


Figure 12. Soil liquid content in HD27 control run (a) and anomaly resulting from implementing
Bruggeman-mixing minerals in the BM experiment (b). The two values in parentheses within the title of
each figure are domain average for the Sahara and Sahel regions.
Over the Sahara region, latent heat flux does not change from HD27 case to BM case, therefore,
the increased net radiative flux ($\sim 3.69\ W/m^2$) in BM compared to HD27 is mainly balanced out
by the enhanced sensible heat flux ($\sim 3.98\ W/m^2$) which requires a larger temperature gradient
between surface and atmosphere. However, there is a very strong negative temperature anomaly
(around $-1K$) in the atmosphere near 700 hPa due to less dust absorption in BM as we discussed
in Figure 10. The strong negative temperature anomaly in the lower atmosphere effectively
increases the vertical temperature gradient. As such, it is not necessary for the land surface
temperature to increase; in fact, it may need to decrease by approximately 0.66 K to achieve the
desired enhancement in sensible heat flux and reach equilibrium.
Additionally, to assess the effectiveness of various dust scattering properties (e.g., HD27, HD09,
and BM) in matching observations of near-surface temperature, we compare the modeled near
surface temperature ($T_{2m}$) with CRU TS observations, which is described in Section 2.6, over the
Sahara and Sahel regions (Table 5). Considering the relatively large inter-model spread of regional
surface air temperature, we compare the Sahara-Sahel regional contrast in surface air temperature
to the CRU rather than comparing their absolute values. Table 5 shows that HD09 and BM improve
the agreement with CRU in Sahara-Sahel temperature contrast compared to HD27, and BM
exhibits the closest agreement with CRU.
**Table 5.** The 19-year (2001~2019) JJA mean 2-meter Air Temperature ($T_{2m}$, unit: K) and their standard
deviation over the 19 years from CRU observations and modeled experiments over the Sahara and Sahel
regions. The 'Contrast' row indicates the $T_{2m}$ regional contrast between the Sahara and the Sahel.

| Region | CRU (K) | HD27 (K) | HD09 (K) | BM (K) |
|---|---|---|---|---|
| Sahara | $305.8 \pm 0.18$ | $306.55 \pm 0.52$ | $306.2 \pm 0.68$ | $305.89 \pm 0.61$ |
| Sahel | $304.1 \pm 0.32$ | $303.19 \pm 0.46$ | $303.87 \pm 0.51$ | $303.89 \pm 0.59$ |
| Contrast | $1.7 \pm 0.5$ | $3.36 \pm 0.98$ | $2.3 \pm 1.19$ | $2.0 \pm 1.2$ |


## 5.3  Impacts on winds and precipitation

To understand the fast circulation and hydrological response resulting from resolving dust
mineralogy, we examine surface wind speed anomalies (Figure 13) and precipitation anomalies
(Figure 14) induced by mineral-resolved dust. We compare mineral-resolved experiments (using
BM as an example) with HD27 and HD09, respectively, to investigate the effects of resolving dust
mineralogy.

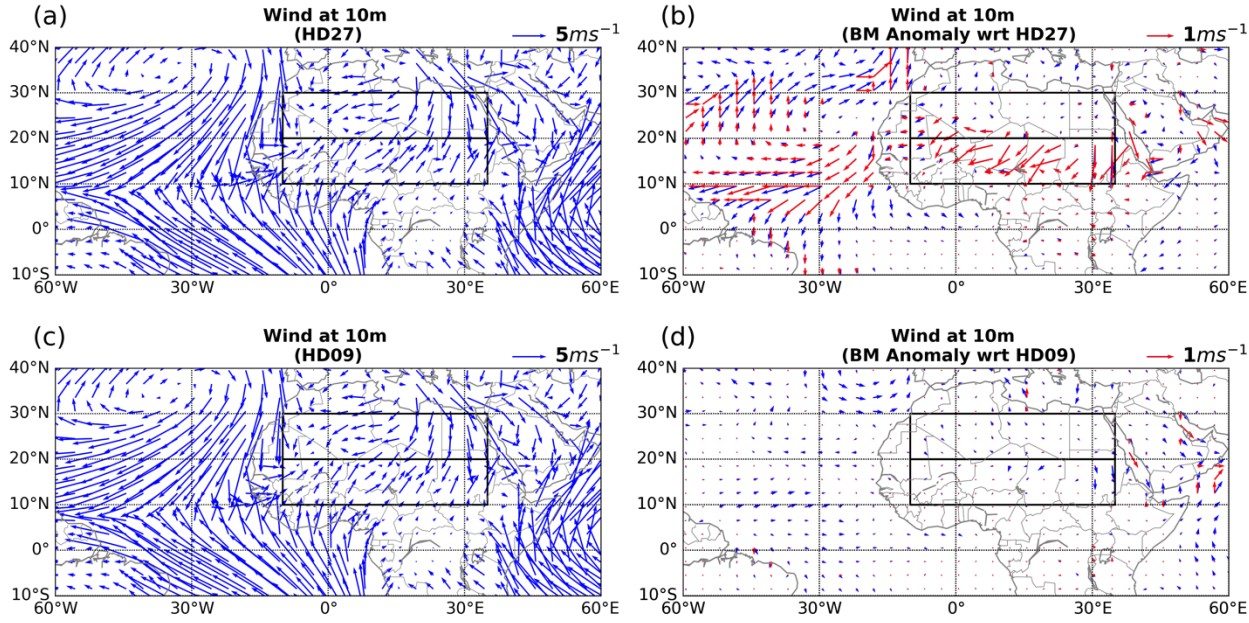


Figure 13. Surface wind at 10-meter from HD27 (a) and HD09 (c) control runs and their anomalies (b) and
(d) resulting from implementing Bruggeman-mixing minerals in the BM experiment. Statistically
significant wind anomalies are highlighted by red arrows.
Global precipitation is higher by 0.017 mm/day for the BM mineral-speciated case compared to
HD27 experiment. This is consistent with the lower SW absorption in the former, given the global
compensation between latent heating associated with precipitation and net radiative cooling (e.g.,
Allen and Ingram, 2002; Samset, 2022). (Global radiative cooling is also compensated, although to a
smaller extent, by the sensible heat flux.)
However, within the Sahel, precipitation for BM is reduced compared to HD27 (Figure 14b), with
weaker onshore flow during the summer monsoon (Figure 13b), displacing West African
precipitation toward the Guinea coast.  This reduction is consistent with several previous
calculations of the fast response calculated with fixed SST (e.g., Stephens et al., 2004; Miller et
al., 2004b; Lau et al., 2009; Jin et al., 2016; Jordan et al., 2018).
Dust absorbs radiation and redistributes heating from the surface to within the dust layer (Miller
and Tegen, 1999; Strong et al., 2015). The heating of the air warms the lower to middle troposphere,
thereby enhancing upward motion. The rising warm air spawns a large-scale onshore flow,
carrying the low-level moist air from the Atlantic to the Sahel, thus enhancing precipitation over
this region (Balkanski et al., 2021). The more scattering mineral-resolved BM and MG dust absorb
less radiation and cause less warming of the atmosphere (Section 4.1), reducing adiabatic cooling
through ascent and Sahel precipitation. The suppressed ascent in BM compared to HD27 is
associated with a reduction both in the wind divergence aloft and in convergence at the surface
(Figure 13b). The reduction in convergence results in northeast wind anomalies at 10-meter over
the Sahel (Figure 13b), which are opposite in direction to the southwest onshore winds of the West
African Monsoon (Figure 13a). The inhibition of onshore winds, bringing less moisture to the
Sahel, is consistent with the reduction of ascent and precipitation over this region (Figure 14a, b).

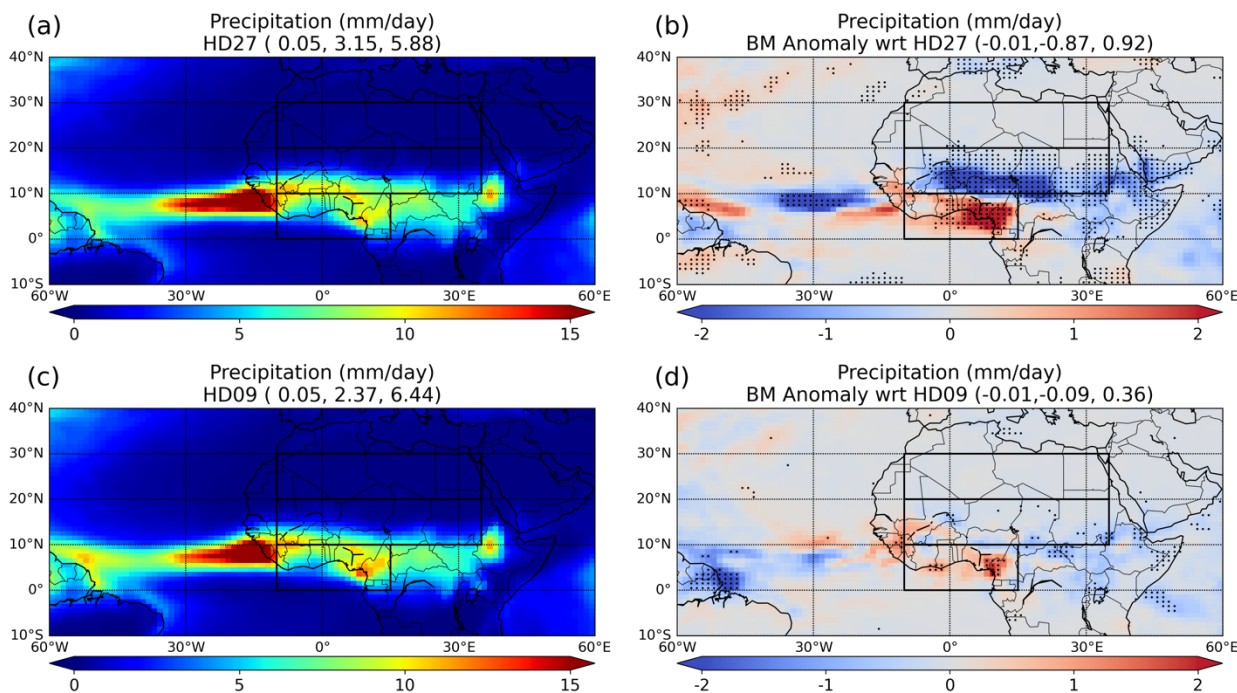


Figure 14. 19-year (2001-2019) JJA mean precipitation from HD27 (a) and HD09 (c) control runs and
anomalies resulting from implementing Bruggeman-mixing minerals with respect to HD27 (b) and with
respect to HD09 (d). The three values in the parenthesis are domain averaged values for the Sahara, Sahel,
and GC regions.
Besides the Sahel, there is a statistically significant positive anomaly (0.92 mm/day) of
precipitation over the region to the south of the Sahel in BM relative to HD27 (Figure 14b). We
will call this region the Guinea Coast (GC: 0-10°N, 10°W-15°E) region. One possible reason for
the increase of precipitation over the GC is that the region is located to the south of the Saharan
dust layer, where the suppression of ascent over the Sahel in BM suppresses the subsidence in the
GC region and, therefore, enhances the precipitation (Guo et al. 2021). Alternatively, the moist
onshore flow that is weakened in BM is subject to greater dilution by the dry desert air, resulting
in reduced moist static energy and buoyancy, limiting convection to the coastal region.
These changes in precipitation have non-negligible effects on soil moisture content in North Africa
due to its moisture-starved environment. The decrease in precipitation over the Sahel in BM leads
to a reduction in soil moisture content. Conversely, the increase of precipitation over the GC leads
to increases of soil moisture (Figure 12). The change in soil moisture content further affects the
partitioning of surface energy fluxes and the efficiency of the latent heat flux, thereby affecting
land surface temperature, as illustrated by Figure 11.
So far, we have been focusing on discussing the impacts of resolving dust mineralogy on winds
and precipitation relative to the HD27 control run. The large discrepancy in optical properties
between HD27 and mineral-resolved dust allows us to better understand how distinct dust
absorption impacts our climate through its distinct radiative effects.
As discussed in section 4.1, HD09 dust is nearly as scattering as mineral-resolved dust but exhibits
smaller regional variability. Section 5.1.2 shows that resolving dust mineralogy does not lead to
statistically significant anomalies on radiation relative to HD09. Consistently, there are no further
statistically significant impacts on winds (Figure 13c, d) and precipitation (Figure 13c, d).
To investigate the effectiveness of various dust scattering properties (e.g., HD27, HD09, BM) in
matching observations of precipitation rate, we compare the modeled precipitation with CRU TS
observations over the Sahara, Sahel and GC regions (Table 6). The greater difference between
HD09, BM and CRU (i.e., HD09 – CRU and BM – CRU) indicate that more scattering HD09 and
BM lead to a larger discrepancy between the modeled precipitation and CRU observations. In
contrast, Balkanski et al. (2021) describes the same balance of increased dust absorption and Sahel
precipitation but find improved agreement with Global Precipitation Climatology Project (GPCP)
data by assuming homogeneous dust containing 3% iron oxides by volume. Contrasts between that
study and ours result from differences between the GPCP and CRU data sets, contrasts in dust
absorption (related to contrasts in the dust size distribution or assumed index of refraction), non-
dust model biases in precipitation or differences between the slow response computed by
Balkanski et al. (2021) and the fast response that we calculate. The fast and slow response even
exhibit differences in the sign of the calculated precipitation anomaly within some regions of the
WAM (Miller and Tegen, 1998; Jordan et al., 2018).
Table 6. Comparison of modeled precipitation rate (PRE, unit: mm/day) with observations from CRU TS
dataset over 2001-2019 JJA. CRU column represents 19-year (2001-2019) JJA mean PRE over the region
as well as 19-year standard deviation (std). HD27 − CRU column shows the 19-year mean PRE difference
between HD27 control run and CRU observations, along with the corresponding std of this 19-year
difference. Similar for HD09 − CRU and BM − CRU.

| Comparison<br><br>Region | CRU<br>(mm/day) | HD27 − CRU<br>(mm/day) | HD09 − CRU<br>(mm/day) | BM − CRU<br>(mm/day) |
|---|---|---|---|---|
| **Sahara** | $0.08 \pm 0.013$ | $-0.03 \pm 0.03$ | $-0.03 \pm 0.07$ | $-0.04 \pm 0.05$ |
| **Sahel** | $2.99 \pm 0.27$ | $0.16 \pm 0.56$ | $-0.62 \pm 0.43$ | $-0.71 \pm 0.41$ |
| **Guinea Coast** | $6.16 \pm 0.49$ | $-0.28 \pm 0.90$ | $0.28 \pm 1.02$ | $0.64 \pm 0.83$ |


# 6 Potential for reducing mineral tracers

Thus far in this study, we have been using 45 mineral tracers in mineral-resolved experiments (i.e.,
VOL, MG, and BM). However, it is important to investigate the potential of reducing the number
of mineral tracers in climate models to lower computational costs. In this section, we take BM as
a reference for providing the best comparisons with CRU temperature and CERES flux
observations, and conduct an experiment named BM-RT to assess the possibility of reducing
mineral tracers in BM. The BM-RT experiment consists of three sub-experiments, namely, BM-
LC, BM-LCRH, and BM-LCRHRG. In each of the three sub-experiments, the number of mineral
tracers is progressively reduced, allowing for an examination of the relative impacts of different
minerals on climate compared to the reference BM.
As discussed in section 4.1, the three clay minerals (i.e., illite, kaolinite, smectite) exhibit similar
optical properties and perform similar functions in climate by hosting hematite. Hence, they can
be combined in their interaction with radiation without significant impacts on climate. In addition,
by lumping the three clay minerals together, the number of mineral tracers can be reduced from
45 in BM (nine types of minerals × five size bins) to 35 (seven types of minerals × five size bins).
Therefore, in the first sub-experiment BM-LC (where 'LC' represents 'Lump Clay minerals'), we
lump together the three clay minerals as one mineral species 'clay433'.
Based upon the C1999 soil mineral composition that we use, externally mixed hematite is mainly
concentrated over the Sahel region (Ginoux et al. 2023, in preparation) and cannot be transported
to remote regions due to its high mass density. Obiso et al. (2023) shows that visible extinction
due to externally mixed hematite is negligible compared to other mineral components including
hematite internally mixed with other minerals. Thus, we further remove external hematite tracers
in the second sub-experiment BM-LCRH (where 'RH' indicates 'Remove externally mixed
Hematite'). The mass fraction of external hematite is combined with internal hematite to ensure
that the total mineral fraction at emission remains equal to one. In this sub-experiment, the number
of mineral tracers is reduced from 35 in BM-LC to 30 in BM-LCRH.
Since there are no known specific impacts of gypsum on climate, we conducted the third sub-
experiment, BM-LCRHRG ('RG' indicates 'Remove Gypsum'), where gypsum was removed. The
mass fraction previously attributed to gypsum at emission, which is very low at the global scale,
was proportionally redistributed among all other minerals. The number of mineral tracers is finally
reduced from 30 in BM-LCRH to 25 in BM-LCRHRG.
We analyze the 19-year (2001-2019) time series of total dust mineral emission before and after
reducing mineral tracers in Figure 15. We observe subtle differences in total mineral emission
between experiments, which arises from the feedback of mineral radiative interactions. However,
these differences are numerically small, and Student's t-test suggests that the time series of the
four experiments are not statistically different. Additionally, the globally averaged DAOD and
SSA of each sub-experiment remains highly similar to those of the reference experiment BM, as
listed in Table 3.

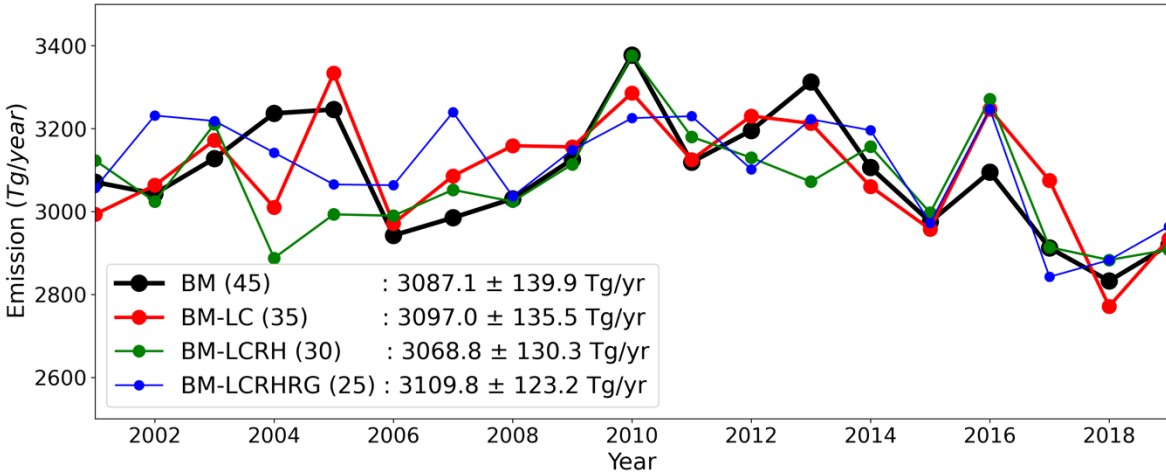

Figure 15. Time series of total dust mineral emission from 2001 to 2019 before and after reducing the number of mineral tracers. The legend displays the following information: 'Experiment name (number of mineral tracers): 19-year averaged total mineral emission ± 19-year standard deviation of total mineral emission'.

Given the highly similar optical properties of minerals before and after reducing mineral tracers, we further investigate their impact on climate. Firstly, we examine the clear-sky flux anomaly of each of the three sub-experiments at TOA and surface relative to the reference experiment BM, as shown in Supplementary Figure S14 for TOA and Supplementary Figure S15 for surface. We only observe a few statistically significant (p-value < 0.05) anomalies over the North Africa, suggesting that the reduction of mineral tracers in the three sub-experiments has a weak impact on radiation.

Furthermore, we investigate the anomaly in temperature profile, surface winds, and precipitation of each of the three sub-experiments relative to the reference experiment BM. The results are presented in Figure S16-S18 in the Supplement. No statistically significant (p-value < 0.05) temperature anomalies (Figure S16) and surface wind anomalies (Figure S17) is observed. Only a few statistically significant anomalies in precipitation are observed in the Supplementary Figure S18. These results suggest that the reduction of mineral tracers in the three sub-experiments has a weak impact on climate.

The results from the BM-RT experiment suggest combining clay minerals and excluding the externally mixed hematite and gypsum to simulate dust impact on radiation. This does not preclude similar conclusion for other impacts of dust on the Earth's climate systems. The removal of mineral tracers reduces the number of mineral tracers from 45 to 25, effectively lowering computational costs without causing statistically significant impacts on simulating climate.

# 7 Conclusions

We simulate the distribution of dust mineralogy (i.e., illite, kaolinite, smectite, hematite, calcite, feldspar, quartz, and gypsum) and activate their interaction with radiation in the GFDL AM4.0 model. Our investigation focuses on the radiative impacts of resolving dust mineralogy on Earth's atmosphere and its fast response of land temperature, surface winds and precipitation.

We set up two baseline homogeneous dust control runs: HD27 and HD09, in which dust mineralogy is considered as temporally and spatially uniform, the former following the standard configuration for the dust optical properties in GFDL AM4.0 and the latter including a more scattering dust. Three experiments with resolved mineralogy are also conducted: VOL, MG, and BG, using three different mixing rules for the internal mixture between hematite and clay minerals (i.e., volume weighted mean, Maxwell Garnett, and Bruggeman). The comparison of dust absorption properties (e.g., SSA) with observation-based results suggests that the homogeneous dust used in the standard GFDL AM4.0 (i.e., HD27) is overly absorptive, Maxwell Garnett and Bruggeman mixing rules are more appropriate than volume weighted mixing rule in calculating optical properties of internal mixtures of hematite and clays. Compared to HD27, the homogeneous dust with reduced hematite content (HD09) and mineral-resolved dust (i.e., MG and BM) exhibit much better agreement with AERONET retrievals and laboratory measurements in terms of dust absorption properties (i.e., SSA). Additionally, resolving dust mineralogy enhances regional variability in dust SSA compared to homogenous dust, further improves the agreement with AERONET, even though it remains lower than the observed variability.

The two homogeneous dust control runs, HD27 and HD09, with distinct dust absorption properties, allow us to investigate the impact of dust mineralogy on Earth's radiation and fast climate response relative to distinct baseline homogeneous dust. In comparison to HD27, resolving mineralogy reduces dust absorption. During JJA, the reduced dust absorption results in a reduction of over 50% in NET downward radiation across the Sahara and approximately 20% over the Sahel at TOA. Additionally, there is a reduction of around 25% in the atmospheric absorption of radiation over the Shahara and around 10% over the Sahel in the atmosphere. The reduced surface absorption of radiation by mineral-resolved dust leads to a temperature decrease of 0.66 K at the land surface across the Sahara and an increase of 0.7 K over the Sahel. The reduced NET downward radiation at TOA, attributed to the less absorption of radiation by mineral-resolved dust, suppresses ascent

and weakens the monsoon inflow from the Gulf of Guinea. This brings less moisture to the Sahel,
which combined with decreased ascent induces a reduction of precipitation. On the other hand,
compared to HD09, resolving dust mineralogy results in dust absorption comparable to that of
HD09 on a global scale. However, when resolving mineralogy, there is an increase in spatial
variation of dust absorption. Additionally, we observe a noticeable change in global distribution
of dust absorption, with more dust absorption distributed in the Southern Hemisphere and lower
dust absorption over Iceland and Taklamakan regions. Nevertheless, the higher spatial variation in
dust absorption does not lead to statistically significant changes in any of the climate aspects
mentioned above. The models with reduced absorption (HD09 and fully resolved mineralogy)
improve the comparison with observations of CERES fluxes and CRU land surface temperature.
We see a slightly better agreement with observations for fully resolved mineralogy than HD09
however it is not statistically significant. As such, when using fixed mineralogical composition,
we recommend using a 0.9% hematite content in volume, which represents the lowest of the three
hematite mixings considered by Balkanski et al. (2007).
Historically, climate models have relied on fixed refractive index to consider dust radiative forcing
starting (IPCC, 2001) with strongly absorptive value based on dust samples in Sahara (Patterson
et al., 1977) to more scattering values after dust absorption could be inferred from satellite and
surface observations (e.g., Sinyuk et al., 2003; Balkanski et al., 2007). With the launch of EMIT
in July 2022 and the expected delivery of a high-resolution map of soil mineralogy in source areas,
dust interactions with radiation in climate models will be calculated directly from the simulated
mineralogical composition (Li et al., 2021). Still, the additional burden of simulating a dozen
minerals may be too prohibitive for large ensemble climate models simulations. In such cases, our
analysis suggests the use of a fixed value providing similar radiative effects as the comprehensive
representation of minerals. However, our recommendation is directed toward the GFDL AM4.0
model with all its uncertainties related to mineral distribution, emission sources, and aerosol
transport. Moreover, incorporating dust mineralogy in models is likely to be important in other
aspects, such as cloud properties, ocean biogeochemistry, air quality and photochemistry. For
studies with resolved mineralogy, we show that the number of mineral tracers can be reduced from
45 to 25 without losing the quality of comparison with observations of CERES fluxes and CRU
surface temperature. Such reduction can be achieved by lumping together clay minerals, removing
external hematite and gypsum. For specific research such as biogeochemistry, it may be necessary
to fully resolve mineralogy to achieve accuracy.
This study has some limitations. First, the soil mineralogy map from C1999 is based on extensive
extrapolation and limited observations. In terms of the need to improve knowledge of soil
mineralogy in dust source regions, the launch in July 2022 of the EMIT instrument operating from
the International Space Station will provide mineral identifications of dust sources using
hyperspectral measurements (Green et al., 2020). The EMIT soil map measurements will improve
resolving dust mineralogy in climate models and advance our understanding of dust's effects in
the Earth system. Second, hematite and goethite are the most common iron oxides present in soils.
However, goethite is not considered in this study because not included in the used soil mineralogy
map. Previous studies suggest that goethite is generally more abundant than hematite, but it is less
absorptive than hematite in the visible spectrum (Formenti et al., 2014). Therefore, the abundance
of iron oxides may be underestimated in this study, which may lead to underestimation of dust
absorption in the SW. A more recent database by Journet et al. (2014) (J2014) includes the
distribution of goethite but it shares many limitations as C1999 (e.g., extensive extrapolation) and
has other major disadvantages, such as numbers of missing soil fractions of some minerals at some
locations. Third, the refractive index of hematite used in our study is close to the upper range of
the values available in literature (Zhang et al., 2015). Hence, the last two limitations,
underestimation of iron oxide content and overestimation of absorption by hematite, may have
compensating effects.
This study, by prescribing SST, calculates only the fast response to the dust DRE, without
including the slow response by the sea surface temperature. This avoids the need to spin-up the
model for decades before reaching new equilibrium but may overestimate the eventual response,
as shown by Miller and Tegen (1998) and Balkanski et al. (2021). This complicates model
evaluation because the observations include the slow response to dust. Variables like precipitation
are especially sensitive to the inclusion of the slow response because prescribed SST experiments
omit the surface energy balance over the ocean. Thus, the surface DRE beneath the aerosol layer,
which is generally negative, is not fully balanced by a fast reduction of evaporation (Miller et al.,
2004a).The addition of the surface balance in the slow response can reverse the sign of the fast
precipitation anomaly (Miller and Tegen, 1998; Jordan et al., 2018). In this study, the increase of
dust scattering (e.g., through consideration of dust mineral composition) generally reduces model
biases for all variables except precipitation. Future works may include satellite-based inventory of
soil mineralogy using fully coupled Earth's system components.

## 8  Competing interests

The contact author has declared that none of the authors has any competing interests.

## 9  Acknowledgement

This research is supported by a collaboration between Princeton University and NOAA GFDL, Cooperative Institute for Modeling the Earth System (CIMES). A portion of this work is funded by the Earth Surface Mineral Dust Source Investigation (EMIT), a NASA Earth Ventures-Instrument (EVI-4) Mission. Carlos Pérez García-Pando and María Gonçalves Ageitos acknowledge support from the European Research Council (ERC) under the Horizon 2020 research and innovation program through the ERC Consolidator Grant FRAGMENT (grant agreement no. 773051), the AXA Research Fund through the AXA Chair on Sand and Dust Storms at the Barcelona Supercomputing Center (BSC), the European Union's Horizon 2020 research and innovation program under grant agreement no. 821205 (FORCeS) and the Department of Research and Universities of the Government of Catalonia via the Research Group Atmospheric Composition (code 2021 SGR 01550). Vincenzo Obiso was supported by the NASA Postdoctoral Program at the NASA Goddard Institute for Space Studies administered by Oak Ridge Associated Universities under contract with NASA (80HQTR21CA005). We acknowledge the CERES EBAF Ed 4.2 data, which were obtained from the NASA Langley Research Center CERES ordering tool at https://ceres.larc.nasa.gov/data/.

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
