# Peer review of "Modeling impacts of dust mineralogy on fast climate response"

_EGUsphere, 2023_

## Referee Comment (RC1)

Review on the "Modeling impacts of dust mineralogy on Earth's Radiation and Climate" by Song et al.

Recent studies suggest growing importance of mineral types in estimating the role of dust to the Earth's climate. However, the impact of soil and dust mineral types is still highly uncertain, while most of global models assume a homogeneous mineral mixture. The present study reports a modeling study of the impact of the mineralogy map to climate system using the GFDL AM4 model. This work starts with the how mineralogy map by Claquin et al. (1999) is implemented to GFDL AM4, and how model experiments and optical properties are setup. The model simulations are evaluated with the observations by di Biagio et al. (2019) and AEROENT data. The present study further examines the impact of dust mineralogy to climate system mainly over North Africa with various parameters such as, radiative fluxes, precipitation, surface temperature, temperature profile, etc.

The method used in the present work is scientifically sounding and it presents several interesting results that might be a beneficial for the potential readers. The paper is well organized and reasonably well written. I would recommend the paper to "minor revision", however I also have suggestions in many places which I would like to see in the revision.

Major comments:

One of the major questions I have is what is the reason for HD2.7% is used as a standard GFDL AM4 models if the hematite fraction is too high? The present study suggests that HD0.9% would be more realistic than HD2.7%. However, the reference study by Balkanski et al. (2007) suggested HD1.5% as the best hematite fraction in the mixture. I would like to see the result of HD1.5% even with a brief result only. Also, it would be good to provide more discussion on this. I would say the differences from 0.9 to 1.5 and to 2.7 % are not negligeable.

The present work needs more background study for the existing research. At least I am aware of a similar recent work by Balkanski et al. (2021, https://doi.org/10.5194/acp-21-11423-2021). How they are compared each other?

Analysis of dust and mineral mass distribution such as horizontal and vertical distribution, and size dependence of distribution is not covered at all in the manuscript, although these are an important step before analyzing climate impact.

The title includes "Earth", however the present paper is limited in North Africa only. There is a brief result in Supplementary Material. I would suggest to expand the work or change the title.

Method section needs more work to make it more clear and specific. I listed a few.

Other comments:

L24-26: It is unclear the reduction of what and is reduced from what?

L101: "Section 2 provides ...." Split the paragraph to a different on.

L110-111: Please be specific if the present work is participating to the AMIP projects.

L137, 138: Change soil map to soil mineralogy map.

L137: Change "to resolve dust (" to "to resolve dust mineralogy (".

L139-141: Please discuss more detail how BFT is implemented to GFDL AM4.0.

L139: Change "disperse" to "fully disperse".

L146-147: The sentence "Moreover, ..." needs to be further elaborated. Please include the difference of goethite and absorption in outside of visible spectrum.

L150-156: This section of internally and externally mixed hematite is confusing and this section needs more improvement in the revision. First, what is the base of partitioning hematite to internal and external mixture in clay? How much of hematite is in internal mixture and external mixture? Second, what is the base of 5% mass fraction threshold for internal and external mixture of the hematite in accretions? Please be specific how Goncalves et al. (2023) used that assumptions. Thirdly, it needs more clear description on how hematite from soil mineral map is used. If the method follows Goncalves et al. (2023), the assumption of 5% is not necessary. Also I wonder how the internal and external mixture assumption is used in other mineral type?

L162: Please specify the meaning of "clay433" here.

L171-174: Optical tables are important for the experiment, and it needs more description here. Or add a sentence that more details will be discussed in Section 4.

L183: SSA and CRI of AERONET is Inversion product. Please specify in the text.

Table 2 BM-RT 2) and 3): Please specify how much of hemitite and gypsum are removed. I am also curious why only LC, RH, and RG are considered among 9 mineral classes?

Figure 1: CRI in longwave length is indistinguishable. I would suggest it to separate to shortwave and longwave.

L269-270: "weighted by solar spectrum"? Please be more specific the meaning.

L280-282: Please discuss why the result in the present paper is different from Balkanski et al. (2007), which concludes that 1.5 % by volume of Hematite using MG mixing gives the best results.

L295-296: Please add discussion for longwave radiation.

L329-330: The sentence is misleading since SSA is affected by both dust size and refractive index (k).

L334-336: Figure S3 and text need more improvement. Please specifically explain why particle size distribution (or SSA) in HD27 is more sensitive than HD09.

L337-338: The sentence needs to be improved in revision too. The previous paragraph and Figure S3 needs to be more clearly described.

L348-351: The present paper concludes that HD09 shows the best agreement with AERONET SSA. The result is not consistent with Balkanski et al. (2007) who found that 1.5 % by volume best matches with AERONET.

L360-361:  Spatial variation of iron oxides content would be one reason for the low spatial variation in modeled SSA. How about model emission, deposition and horizontal- and vertical-transport?

L379: Linear averaging is not a correct way. Please specify if SSA interpolated to 550nm with Angstrom Exponent.

L380-381: "Figure 2 shows..." is not necessary. Delete the sentence.

L381-382: The sentence is unclear. Why MB is not representative if MB and BM are similar each other?

L397-398: The sentence about EMIT would belong to discussion.

There are five figures (Figures  4-8) in Section 5.1. However there is no table. I would suggest to provide a table that summarizing DRF.

L422-423: North Africa is hotspot. However global mean estimation is also important to examine the impact of dust mineralogy.

L451-453: Please provide actual number for 25% and 10% reductions. Also it is unclear the change is same for three mineralogy resolved tests or they are given in range. The same applied to other figures 4, 5, 7.

L455: Global estimate is an important result of the present study even if the magnitude is not as large as North Africa. I would suggest including a table for global radiative forcing. How about other major dust sources such as Asia and Australia?

L470-472: The sentence about EMIT is not result of this work, it belongs to discussion.

L473-482: The study argues that HD09 is better than HD27 in the comparison with CERES radiation flux observations. However it is unclear if the better agreement in HD09 is the correction of the model or something else. (1) The CERES observation includes many factors including many aerosol types such as sea-salt, biomass, burning, anthropogenic, and volcano. (2) Many factors are involved in dust radiation flux calculation, e.g., dust amount, optics table, size distribution, etc. I think one can make better agreement with CERES by modify one these fields.

L481-482: Please be specific about "better".

L515: From Figure 9, Maybe it is worthy to point out that the difference between HD27 and HD09 is about +1 deg K.

L555-560: Please make the two sentence more clear.

L607-610: I wonder if the result is based on observation or speculation. If authors have seen during analysis it can be written more firmly.

Section 6: The authors argue that reducing mineral types do not have significant impact on radiative forcing. However it should be pointed the mineralogy also have importance on geochemical cycles and cloud nucleation.

L665: I still do not understanding what actually means 'clay433'. Could it be more human understandable?

L718-720: The finding that the dust in the standard AM4.0 (HD27) is too absorbing is quite surprising. I would like to see more this finding. What is the background of choosing HD27 other than lower iron-oxides. Is it based on studies or inherited from OPAC type of table?

The result is also soil mineral map dependent. The iron oxides content would be different from Claquine et al. to EMIT. This needs to be included.

L471-473: Again, I wonder if this recomendation is also consistent with Balkanski et al. (2007 and 2021).

L760-770: The present study needs to further esmitate or discuss the uncertainty by missing Goethite in ironoxides.

---

## Author Comment (AC1)

**Response to RC1**

Recent studies suggest growing importance of mineral types in estimating the role of dust to the Earth's climate. However, the impact of soil and dust mineral types is still highly uncertain, while most of global models assume a homogeneous mineral mixture. The present study reports a modeling study of the impact of the mineralogy map to climate system using the GFDL AM4 model. This work starts with the how mineralogy map by Claquin et al. (1999) is implemented to GFDL AM4, and how model experiments and optical properties are setup. The model simulations are evaluated with the observations by di Biagio et al. (2019) and AEROENT data. The present study further examines the impact of dust mineralogy to climate system mainly over North Africa with various parameters such as, radiative fluxes, precipitation, surface temperature, temperature profile, etc.

The method used in the present work is scientifically sounding and it presents several interesting results that might be a beneficial for the potential readers. The paper is well organized and reasonably well written. I would recommend the paper to "minor revision", however I also have suggestions in many places which I would like to see in the revision.

Major comments:

One of the major questions I have is what is the reason for HD2.7% is used as a standard GFDL AM4 models if the hematite fraction is too high? The present study suggests that HD0.9% would be more realistic than HD2.7%. However, the reference study by Balkanski et al. (2007) suggested HD1.5% as the best hematite fraction in the mixture. I would like to see the result of HD1.5% even with a brief result only. Also, it would be good to provide more discussion on this. I would say the differences from 0.9 to 1.5 and to 2.7 % are not negligeable.

**Reply:** Thanks for the comments.
The reason of using HD2.7% in the standard GFDL AM4 model:
Fixing 2.7% hematite content for dust particles was decided during the development of the previous GFDL Climate Model CM3 (Donner et al., 2011) when it was found that dust absorption was unrealistically high (by a factor 3) in CM2 (Delworth et al., 2006) compared to AERONET observations (Balkanski et al., 2007). The conjunction in CM3 of a sharp decrease of black carbon (strong aerosol absorber) with a new emission inventory and the switch to more scattering dust had a negative effect on precipitation bias, and late 20th century warming (see Donner et al., 2011 for details). To limit this bias, selecting 2.7% hematite was adopted.
*Reference: Donner, L.J., Wyman, B.L., Hemler, R.S., Horowitz, L.W., Ming, Y., Zhao, M., Golaz, J.C., Ginoux, P., Lin, S.J., Schwarzkopf, M.D. and Austin, J., 2011. The dynamical core, physical*

*parameterizations, and basic simulation characteristics of the atmospheric component AM3 of the GFDL global coupled model CM3. Journal of Climate, 24(13), pp.3484-3519.*

The fractional content of hematite in dust aerosols along with its complex refractive index (CRI) determines dust absorptivity (e.g., SSA) in the shortwave spectrum. As we can see from Figure 5 (a) in the manuscript, AERONET retrieved dust SSA (0.938) is higher than both HD2.7% (0.853) and HD0.9% (0.929), given out choice of CRI (Figure1). This suggests that AERONET retrievals imply an equivalent or even lower hematite content in dust aerosols than 0.9%. Considering this, the SSA of HD1.5% would fall between the values for HD2.7% and HD0.9%, aligning better with AERONET than HD2.7% but not as well as HD0.9%. Therefore, HD1.5% is not used in this study.

In addition, the most appropriate suggested hematite content may vary across studies due to the significant uncertainty in the CRI of hematite. A higher hematite content, when paired with a lower imaginary part of the RI, would be equivalent in dust SSA to a combination of lower hematite content with a higher imaginary part of the RI. For example, Balkanski et al. (2007) took their hematite CRI from Bedidi and Cervelle (1994). This CRI has a lower imaginary part, corresponding to less absorption, than the value we choose following Scanza et al. (2015). (See our Figure 1 and Figure 4 of Obiso et al., 2024). This is one reason that we arrive at a lower optimal fraction of hematite than Balkanski et al. (2007).

The present work needs more background study for the existing research. At least I am aware of a similar recent work by Balkanski et al. (2021, https://doi.org/10.5194/acp-21- 11423-2021). How they are compared each other?

Reply: Thanks for the suggestion.
Balkanski et al. (2021) mainly focused on investigating the enhancement of precipitation over the Sahel induced by absorbing dust, using a fully coupled model compared to a no-dust condition. Our study uses an atmospheric model with prescribed SST and compares a mineralogy-resolving case with mineralogy non-resolving case. Nevertheless, our findings align consistently with theirs, indicating that more absorptive dust enhances precipitation over the Sahel region. Therefore, we have cited this paper to support our results regarding the enhancement of precipitation due to dust absorption.

Analysis of dust and mineral mass distribution such as horizontal and vertical distribution, and size dependence of distribution is not covered at all in the manuscript, although these are an important step before analyzing climate impact.

Reply: Thanks for the great comment.

We have included an analysis of the global distribution of dust optical depth (DAOD) in Figure 2 to provide a proxy of the vertically integrated dust distribution in the main text. The size dependence of global dust (or mineral) mass has been added in the section S2 in the supplement. We have added the discussion about the added figures as:

'In addition to the globally averaged dust properties listed in Table 3, we illustrate the global distribution of DAOD (Figure 2) and the distribution of global dust mass across 5 size bins (Figure S3 in the Supplement) for the three experiments: before (e.g., HM27 and HD09) and after (e.g., BM) resolving mineralogy. The global dust size distribution across the 5 size bins remains largely unchanged across experiments. Besides the subtle difference (~10%) in global mean DAOD across the three experiments as listed in Table 3, the global distribution of DAOD responds differently in HD09 and BM. Compared to HD27, reducing hematite content in HD09 generally decreases DAOD, except over the Sahel region. In contrast, resolving mineralogy decreases DAOD over the Sahara region while increasing DAOD over the Sahel and Asia regions. The reduction in DAOD over the Sahara region further contributes to the decrease in dust absorption over the region, primarily attributed to the change in dust optical properties, such as the enhancement in dust SSA. The indistinct variation in DAOD across different experiments results from the feedback of dust interactions with radiation (Miller et al.,2004; Pérez et al., 2006; Miller et al., 2014), which is influenced by the distinct scattering properties of dust aerosols in each experiment as shown in Table 3.'

[Figure]

Figure 2 in the main text. The global distribution of dust optical depth (DAOD) for HD27, and the difference in DAOD between HD09 and BM from HD27. The global mean DAOD values ($\overline{DAOD}$) of each experiment are shown in Table 3 in the main text.

[Figure]

Figure S4 in the Supplement (bin1: 0.2 – 2 $\mu m$, bin2: 2 - 3.6 $\mu m$, bin3: 3.6 – 6 $\mu m$, bin4: 6 – 12 $\mu m$, bin5: 12 - 20 $\mu m$). The dust load fraction in each size bin is analyzed for two homogeneous dust experiments (e.g., HD27 and HD09) and a mineral-resolved experiment (e.g., BM). These fractions are calculated based on the 19-year (2001-2019) global dust load.

The title includes "Earth", however the present paper is limited in North Africa only. There is a brief result in Supplementary Material. I would suggest to expand the work or change the title.

Reply: Thanks for the comment. The title is changed to 'Modeling impacts of dust mineralogy on fast climate response'
We think this is a global study using a climate model. The dust optical properties (e.g., SSA) are compared over global scope. The impacts of dust mineralogy on climate are also investigated in global scope (in supplementary). However, the statistically significant anomalies primarily occur over North African in JJA, therefore, we focus on North Africa in section 5 of the main text to discuss the impacts of dust mineralogy.

Method section needs more work to make it more clear and specific. I listed a few.

Other comments:

L24-26: It is unclear the reduction of what and is reduced from what?

Reply: Thanks. We clarified the statement by saying that 'Over the 19-year (from 2001 to 2019) modeled period during JJA (June-July-August), resolving dust mineralogy leads to a reduction of over 50% in net downward radiation across the Sahara and approximately 20% over the Sahel at top of atmosphere (TOA) compared to the baseline bulk dust model version.'

L101: "Section 2 provides ...." Split the paragraph to a different on.

Reply: Thanks. Done

L110-111: Please be specific if the present work is participating to the AMIP projects.

Reply: Sorry for the confusion. This work is not participating in the AMIP projects. We have revised the sentence to avoid confusion.
'We conduct a series of experiments with GFDL AM4.0 (Zhao et al. 2018a, b) over the period 2001-2019 using the AMIP protocol, meaning sea surface temperature (SST) and sea-ice are imposed based upon average monthly observations (see Gates, 1992 for details) .'

L137, 138: Change soil map to soil mineralogy map.

Reply:  Done

L137: Change "to resolve dust (" to "to resolve dust mineralogy (".

Reply: Done

L139-141: Please discuss more detail how BFT is implemented to GFDL AM4.0.

Reply: Thank for the comment. We have included a brief description of implementing dust mineralogy following BFT as follows.
'The soil map is based on soil analyses that are usually done after wet sieving, which disperse mineral aggregates into small particles. This dispersal is particularly relevant for the phyllosilicates, typically found in the form of aggregates in soils. They are detected in the atmosphere with higher proportions at coarser (silt) sizes than those reported in the soil maps (Perlwitz et al., 2015b; Perez Garcia-Pando et al., 2016). These recent studies also show that the Brittle Fragmentation Theory (BFT; Kok, 2011) represents a practical framework to generate the emitted particle size distribution based on the dispersed soil PSD, which facilitates the utilization of soil mineralogy maps. In our simulations, we employ BFT to reconstruct the mineral aggregates emitted from the original undispersed soils, following the methods described in Gonçalves Ageitos et al., 2023.'

L139: Change "disperse" to "fully disperse".

Reply: Done

L146-147: The sentence "Moreover, ..." needs to be further elaborated. Please include the difference of goethite and absorption in outside of visible spectrum.

Reply: Thanks! We have provided further elaboration on the statement and included a reference to support it.

'Goethite and hematite are the two major types of iron oxides present in soils. Goethite is less absorptive than hematite and is not resolved in C1999. So, iron oxides are represented by hematite in this study. Hematite has larger density than other minerals, so that hematite deposits more quickly and is not able to be transported to remote regions when not aggregated or internally mixed with lighter clay minerals. Moreover, among the minerals considered here, hematite is the strongest absorber at ultraviolet (UV) and visible wavelengths, while it does not have noticeable absorption at infrared wavelengths (IR) compared to other minerals (Sokolik and Toon, 1999).'

L150-156: This section of internally and externally mixed hematite is confusing and this section needs more improvement in the revision. First, what is the base of partitioning hematite to internal and external mixture in clay? How much of hematite is in internal mixture and external mixture? Second, what is the base of 5% mass fraction threshold for internal and external mixture of the hematite in accretions? Please be specific how Goncalves et al. (2023) used that assumptions. Thirdly, it needs more clear description on how hematite from soil mineral map is used. If the method follows Goncalves et al. (2023), the assumption of 5% is not necessary. Also I wonder how the internal and external mixture assumption is used in other mineral type?

Reply: Thanks for the comments.
I have modified this part by adding more references and clarifying the way of partitioning hematite into internally mixed and externally mixed portions.

'we partition hematite into two portions by defining two sets of tracers: one set of tracers carries the mass of the hematite that constitutes small accretions in clay minerals (i.e., internally mixed with clay minerals), are allowed to be up to 5 % of the masses of their host minerals at emission (Perlwitz et al., 2015; Gonçalves Ageitos et al., 2023). Given the low fractional mass of hematite compared to their host minerals, we assume that these accretions do not change the density of their host particles. These internally mixed accretions form the largest fraction of the emitted hematite. Another smaller set of tracers carries the mass of the remaining fraction of hematite, which is considered to be externally mixed with the other minerals, including the internal mixtures of hematite with clay. All other minerals are considered to be externally mixed.'

L162: Please specify the meaning of "clay433" here.

Reply: Thanks. We have specified the meaning of clay433 in the manuscript.
'The clay433 represents a mixed mineral comprising three clay minerals: illite, kaolinite and smectite, with mass fraction of 40%, 30%, and 30%, respectively (see detailed descriptions in Supplementary Section S1).'

L171-174: Optical tables are important for the experiment, and it needs more description here. Or add a sentence that more details will be discussed in Section 4.

Reply: Thanks for the suggestion. I have added a sentence stating, 'More details about optical properties of minerals will be discussed in Section 4.'

L183: SSA and CRI of AERONET is Inversion product. Please specify in the text.

Reply: Thanks. Done

Table 2 BM-RT 2) and 3): Please specify how much of hematite and gypsum are removed. I am also curious why only LC, RH, and RG are considered among 9 mineral classes?

Reply: Thanks for the question.
I modified the Table 2 to make it clear that the tracers of hematite and gypsum are removed in BM-LCRH and BM-LCRHRG, respectively.
The reasons for considering LC, RH and RG are stated in Section 6 in details (see below). Other minerals are important to climate in various ways as discussed in the introduction. For example, internally mixed hematite is important for SW absorption of dust, calcite is important for chemistry, feldspar is important for ice nucleation, quartz is important for LW absorption.
For LC: 'As discussed in section 4.1, the three clay minerals (i.e., illite, kaolinite, smectite) exhibit similar optical properties and perform similar functions in climate by hosting hematite. Hence, they can be combined in their interaction with radiation without significant impacts on climate.'
For RH: 'Based upon the C1999 soil mineral composition that we use, externally mixed hematite is mainly concentrated over the Sahel region (Ginoux et al. 2023, in preparation) and cannot be transported to remote regions due to its high mass density. Obiso et al. (2023) shows that visible extinction due to externally mixed hematite is negligible compared to other mineral components including hematite internally mixed with other minerals. Thus, we further remove external hematite tracers in the second sub-experiment BM-LCRH (where 'RH' indicates 'Remove externally mixed Hematite').'
For RG: 'Since there are no known specific impacts of gypsum on climate, we conducted the third sub-experiment, BM-LCRHRG ('RG' indicates 'Remove Gypsum'), where gypsum was removed.'

Figure 1: CRI in longwave length is indistinguishable. I would suggest it to separate to shortwave and longwave.

Reply: Thanks for the comment. We have revised the figure to improve the clarity in LW.

[Figure]

L269-270: "weighted by solar spectrum"? Please be more specific the meaning.

Reply: Thanks for the comment. We have specified the solar spectrum as follows.
'Additionally, the spectrally averaged DAOD and SSA are always weighted by the TOA solar radiation intensity at the corresponding wavelengths, peaking around 0.50 μm, as shown in Eq. (1) and Eq. (2).

$$\overline{DAOD} = \frac{\int_{\lambda 1}^{\lambda 2} DAOD(\lambda)\, B(\lambda)\, d\lambda}{\int_{\lambda 1}^{\lambda 2} B(\lambda)\, d\lambda} \qquad \text{Eq. (1)}$$

$$\overline{SSA} = \frac{\int_{\lambda 1}^{\lambda 2} SSA(\lambda)\, DAOD(\lambda)\, B(\lambda)\, d\lambda}{\int_{\lambda 1}^{\lambda 2} B(\lambda)\, DAOD(\lambda)\, d\lambda} \qquad \text{Eq. (2)}$$

Where $B(\lambda)$ describes the solar radiation energy intensity, which can be calculated by means of the Planck's function B(T,λ), using the temperature of the Sun (T = 5800 K).'

---

## Author Comment (AC2)

**Response to RC2**

This article presents a comprehensive analysis of the effect on climate of specifying dust mineralogy in the GFDL AM4.0 model. This is a topic of great current interest with the success of the Earth Surface Mineral Dust Source (EMIT) mission. Overall the paper is well written and presents interesting and topical results. I have a few important concerns but am hopeful that the paper will be suitable for acceptance after major revisions.

My main concern is with the framing of the results. Specifically, the authors mostly compare the HD27 run with three mineralogy-resolved runs with different mixing rules (VOL, MG, and BM). The problem is that not one but two important things differ between HD27 and these three runs: (1) the mineralogy is resolved only for the VOL, MG, and BM runs and not for HD27, and (2) the absorption in the mineralogy-resolved runs is only about a third of that in HD27, which is clearly too absorbing in comparison to AERONET (Fig. 3) and other data. So the large differences in radiative fluxes (Figs. 4-6, 8), temperature (Fig. 10), soil moisture (Fig. 11), wind (Fig. 12), and precipitation (Fig. 13) between HD27 and the mineralogy-resolved runs are predominantly because of the very large difference in atmospheric absorption between these runs, whereas mineralogy being resolved or not plays only a minor role. The authors acknowledge this in various places (e.g., Fig. 7) but still commonly present their results as being due to resolving mineralogy, which I think is not accurate and misleading to the reader. Examples include:

- Line 21 – 23 (abstract): "Resolving dust mineralogy reduces dust absorption and results in improved agreement with observation-based single scattering albedo (SSA), radiative fluxes from CERES (the Clouds and the Earth's Radiant Energy System), and land surface temperature from CRU (Climatic Research Unit), compared to the baseline bulk dust model version." But these improvements are driven predominantly by reducing dust absorption (i.e., this happens in going from HD27 to HD09), which is set to an unrealistically high value in the control run, to a more reasonable value, irrespective of whether or not the mineralogy is resolved.

- Line 25-26 (abstract): "[Resolving dust mineralogy] leads to a reduction of over 50% in net downward radiation across the Sahara and approximately 20% over the Sahel at top of atmosphere (TOA)." This again is primarily due to the reduction in dust absorption, not because of resolving dust mineralogy.

- Line 437-9: "resolving mineralogy turns out to induce substantial decrease in NET flux at TOA, with a more than 50% negative anomaly over the Sahara and around a 20% negative anomaly over the Sahel (see values in parentheses in Figure 4 i-l)." Here again the attribution to resolving mineralogy seems inaccurate. The authors obtain similar results simply by scaling down the unrealistically high absorption in HD27 to a more reasonable value (in HD09). The authors note this in the next sentence but still leaves the reader with the impression that this is due to resolving mineralogy.

The key issue in all these examples (and various other ones throughout the paper) is that the authors present the reduction in dust absorption as being a result of resolving dust mineralogy, but this is artificial because the absorption was set to a very high value and we've known for a

while know that absorption is much lower than used in the HD27 run (e.g., Balkanski et al. ACP, 2007; Di Biagio et al., ACP, 2019; Adebiyi et al., Comm Earth & Env, 2023).

To address this, I recommend that the authors first have a section where they analyze the effects of dust absorption by comparing HD27 with HD09 – since HD09 has more realistic absorption that is similar to that in the mineralogy-resolved runs - and only then analyze the effects of resolving dust mineralogy by comparing HD09 with the three mineralogy-resolved runs. This would separate the effects of reducing absorption to a more realistic value from the actual effect of resolving mineralogy, thereby preventing the reader from misunderstanding the attribution of the results, and would get the paper's main points across more clearly and accurately.

Relatedly, I think figures 4-6 should be removed because the differences in fluxes here are primarily because of the very large difference in absorption between HD27 and the mineralogy-resolved cases, not because of resolving mineralogy per se. So these figures can cause readers to draw the incorrect conclusion that these flux differences are due to resolving mineralogy. Instead, the authors could show figures comparing HD27 to HD09 (i.e., expand the current Figure 7 into three figures similar to Figures 4-6).

**Reply:** Thank you so much for the great comment and suggestion.
In comparison to HD27, resolving mineralogy reduces dust absorption and enhances the spatial variation in dust SSA. Whereas, compared to HD09, resolving mineralogy leads to comparable dust absorption from a global mean perspective, but increases the spatial variation in dust absorption. In fact, the impact of resolving mineralogy depends on the choice of the homogeneous dust control run. We agree that, compared to HD27, the significant anomaly on radiation and climate after resolving mineralogy is mainly due to the reduction of dust absorption. Therefore, we should not simply confuse the impact of reducing absorption and resolving mineralogy. We clarify the point in the manuscript.

For example:
- (Abstract) We modify the sentence as 'Resolving mineralogy reduces dust absorption compared to the homogeneous dust used in the standard GFDL AM4.0 model that assumes a globally uniform hematite volume content of 2.7% (HD27). The reduction in dust absorption results in improved agreement with observation-based single scattering albedo (SSA)… To isolate the effect of reduced absorption, compared to resolving spatial and temporal mineralogy, we carry out a simulation where the hematite volume content of homogeneous dust is reduced from 2.7% to 0.9% (HD09). The dust absorption (e.g., single scattering albedo) of HD09 is comparable to that of the mineralogically speciated model on a global mean scale, albeit with a lower spatial variation that arise solely from particle size. Comparison of the two models indicates that the spatial inhomogeneity in dust absorption resulting from resolving mineralogy does not have significant impacts on Earth's radiation and climate, provided there is a similar level of dust absorption on a global mean scale before and after resolving dust mineralogy.'.
- At the beginning of Section 5, we add an explanation for the choice of two control runs in the study and explicitly list the factors causing the differences after resolving mineralogy as follows.

(Section 5) 'To investigate the impacts of resolving dust mineralogy on climate, we need to compare modeled results in mineral-resolved experiments to the baseline homogeneous dust (i.e., non-resolved mineralogy) control run. As a result, the significance of the impacts depends on the selection of the baseline homogeneous dust experiment. In this section, we investigate the impacts of resolving dust mineralogy compared to two baseline homogeneous dust experiments. One baseline experiment is the homogeneous dust used in the standard GFDL AM4.0, in which dust mineralogy is assumed to be temporally and spatially uniform (i.e., non-resolved), with a volume fraction of 2.7% hematite (HD27). The impacts of resolving dust mineralogy, compared to the baseline HD27, can be attributed to two factors: 1) the reduction in dust absorption after resolving dust mineralogy in comparison to HD27 as discussed in Section 4, and 2) spatial and temporal variations in dust scattering properties induced by inhomogeneity in dust mineralogy. The other baseline experiment is the homogeneous dust, with a volume fraction of 0.9% hematite (HD09). Given the comparable global mean dust scattering properties (e.g., $\overline{SSA}$) between the control run HD09 and mineral-resolved experiments (e.g., MG and BM), the impacts of resolving dust mineralogy, compared to baseline HD09, is solely attributed to the inhomogeneity in dust scattering properties induced by resolving dust mineralogy.'

- We have added a separate section (5.1.2) to discuss the impacts of resolving mineralogy on Clear-sky Radiative Fluxes relative to HD09

In addition, we would like to emphasize that HD27 represents the homogeneous dust that has been used in the standard GFDL models since the development of GFLD CM3. Therefore, from a model development perspective, it is important to investigate the impacts of resolving mineralogy compared to HD27. This comparison serves as a reference for all other climate models utilizing homogeneous dust with excessive absorption, as there is no guarantee that the dust absorption (e.g., SSA) in their climate models will remain unchanged after resolving dust mineralogy. Additionally, this study provides a valid reference for GFDL to reduce dust absorption in their models. Therefore, in the manuscript, we would like to keep the impacts on Earth's radiation and Climate after resolving mineralogy compared to HD27. However, we explicitly explain that, compared to HD27, the impacts of resolving mineralogy are due to two factors: 1) dust absorption reduction and 2) the spatial and temporal inhomogeneity in dust absorption properties induced by resolving mineralogy. To investigate the impacts of resolving mineralogy, which are solely attributed to the second factor (inhomogeneity in dust absorption properties), we have added a separate Section 5.1.2 to discuss the comparison between mineral-resolved experiment (e.g., BM) and HD09.

Another concern is that the authors assess the effect of mineralogy on absorption and SSA using comparisons against data, but do not sufficiently consider the effect of particle size in these comparisons.

- For instance, on line 346 they state that the high standard deviation of SSA in AERONET data is due to spatial variability in mineralogy but this could also be due to spatial variability in the dust size distribution. As far as I can tell, the authors are not able to distinguish between those two sources of SSA variability in the real world.

- Similarly, line 361 states "The underestimation of regional SSA contrast in AM4.0 suggests the need for a higher regional contrast in iron oxides content". But a higher

regional contrast in the dust size distribution would similarly increase variability in SSA.

More clearly addressing the effect of model errors in the size distribution is important since the size distribution affects most/all of the results presented in the paper and because it seems that (super) coarse dust accounts for a lot of the absorption over North Africa (e.g., Ryder et al., ACP, 2019, Figure 7c) and it's unclear to what extent this is represented in the current paper.

To address this, I recommend that the authors are (1) clearer about the exact size distribution they use in their simulations (see comment below) and (2) discuss clearly that errors in both size distribution and mineralogy can cause the disagreements against the SSA AERONET data.

**Reply:** Thank you so much for the insightful comments.
(1) We have included the exact size ranges of each bin in the manuscript and explain that the choice of the emitted fraction is based on Kok et al. (2011). 'Dust size is represented by five bins with diameter ranging from 0.2 $\mu m$ to 20$\mu m$ (bin1: 0.2 – 2 $\mu m$, bin2: 2 - 3.6 $\mu m$, bin3: 3.6 – 6 $\mu m$, bin4: 6 – 12 $\mu m$, bin5: 12 - 20 $\mu m$). The corresponding source fractions have been updated from 0.1, 0.225, 0.225, 0.225 and 0.225 to values of 0.04, 0.14, 0.19, 0.49, and 0.14 for the five bins. These updated source functions allocate more fraction to coarser size bins, following the suggested Brittle Fragmentation Theory (BFT) as proposed by Kok et al. (2011).'
(2) We have added a discussion to clearly state that errors in both size distribution and mineralogy can cause the disagreements against the SSA AERONET data. We changed the sentence from 'The underestimation of regional SSA contrast in AM4.0 suggests the need for a higher regional contrast in iron oxides content.' to 'The significant underestimation of regional SSA contrast in AM4.0, even after accounting for mineralogy, implies that something important is still missing in the model. For instance: 1) in reality, the regional contrast in iron oxides content may be higher than that resulting from the soil map used in this study, and 2) the dust size, which determines SSA along with mineralogy, may not be well represented in the model. Specifically, the model may underestimate the regional contrast in dust size distribution, and 3) spatial variation of dust shape, which is not taken into account in the model.'

My final major more substantial comment is that I find it difficult to pinpoint exactly what we have learned from this paper that we didn't already know from previous studies of the effect of dust absorption on radiative fluxes and the monsoon over Northern Africa (e.g., Balkanski et al., ACP, 2021) and the effect of resolving mineralogy (e.g., Scanza et al., 2015; Perlwitz et al., 2015; Li et al., 2021; Li et al., 2021; Gonçalves Ageitos et al., 2023; Obiso et al., 2023). I would suggest spelling this out more clearly in (especially) your conclusions section.

**Reply:** Thanks for your comment.
To spell out the uniqueness of this study in comparison with previous studies, we have undertaken the following efforts:
1) In the introduction, we added: 'Following the recent launch of the Earth Surface Mineral Dust Source Investigation (EMIT) instrument specifically designed to retrieve global distribution of dust mineralogy over dust sources (Green et al., 2020), there have been coordinated efforts to represent dust mineralogy *and investigate DRE of mineral-*

*speciated dust* in climate models, in particular in Li et al. (2021), Gonçalves Ageitos et al. (2023), and Obiso et al. (2023). *However, to the best of our knowledge, there have been no studies investigating the fast climate impact of dust while accounting for its mineral speciation. Our work contributes to these efforts by incorporating dust mineralogy into the GFDL models, and it is distinguished extending its investigation to the fast climate response of mineral-speciated dust.*'

In the conclusion, we added a paragraph to reiterate the distinctive aspect of this study: 'Historically, climate models have relied on fixed refractive index to consider dust radiative forcing starting (IPCC, 2001) with strongly absorptive value based on dust samples in Sahara (Patterson et al., 1977) to more scattering values after dust absorption could be inferred from satellite and surface observations (e.g., Sinyuk et al., 2003; Balkanski et al., 2007). With the launch of EMIT in July 2022 and the expected delivery of a high-resolution map of soil mineralogy in source areas, dust interactions with radiation in climate models will be calculated directly from the simulated mineralogical composition (Li et al., 2021). Still, the additional burden of simulating a dozen minerals may be too prohibitive for large ensemble climate models simulations. In such cases, our analysis suggests the use of a fixed value providing similar radiative effects as the comprehensive representation of minerals.'

2) Previous studies investigating the effect of dust absorption on radiative fluxes and monsoon (e.g., Balkanski et al. 2021) assume globally uniform dust mineral composition. They neglect regional variations in the dust DRE resulting from mineral variations. However, this study addresses the effect of mineral variations. In addition, Balkanski et al. (2021) considers the 'slow' response by using fully coupled model, whereas this study considers the 'fast' response. In Section 5.3, we included the comparison of our fast precipitation response with Balkanski et al. (2021) as follows: '*In contrast, Balkanski et al. (2021) describes the same balance of increased dust absorption and Sahel precipitation but find improved agreement with Global Precipitation Climatology Project (GPCP) data by assuming homogeneous dust containing 3% iron oxides by volume. Contrasts between that study and ours result from differences between the GPCP and CRU data sets, contrasts in dust absorption (related to contrasts in the dust size distribution or assumed index of refraction), non-dust model biases in precipitation or differences between the slow response computed by Balkanski et al. or the fast response that we calculate. The fast and slow response even exhibit differences in the sign of the calculated precipitation anomaly within some regions of the WAM (Miller and Tegen, 1998; Jordan et al., 2018).*'

Other comments:

- There is some confusing wording in the paper when it comes to describing the differences between the different runs, which is often the difference in the difference between radiation calls with and without dust. For instance, I think line 25 (Abstract) should read "it leads to a reduction of over 50% in the dust effect on net downward radiation" (underlined words were added). There are similar issues with wording in the conclusions section (e.g., lines 728-730)

Reply: Thanks for the comment, but we will retain our original wording for the following reasons.

In the third row of Figure 6 in the manuscript, we show the net downward flux at TOA, rather than the effect of dust on the downward flux at TOA. Those are distinct concepts because assessing the effect of dust on the radiation involves comparing both the with-dust and no-dust cases and calculating the difference between them. While, in this study, we only show the radiation in with-dust cases.

We can see that compared to the homogeneous dust control run (i.e., HD27, Fig 6i), mineral-resolved experiment (Fig 6l) reduces net downward radiation at TOA by more than 50% over the Sahara.

- In section 2.1, what are the exact size ranges of the bins? And what is the choice of the emitted fraction per bin based on?

Reply: Thanks for the comment.
We add the exact size range of each bin in the manuscript and explain that the choice of the emitted fraction is based on Kok et al. (2011). The manuscript is revised as follows.
'Dust size is represented by five bins with diameter ranging from 0.2 $\mu m$ to 20$\mu m$ (bin1: 0.2 – 2 $\mu m$, bin2: 2 - 3.6 $\mu m$, bin3: 3.6 – 6 $\mu m$, bin4: 6 – 12 $\mu m$, bin5: 12 - 20 $\mu m$). The corresponding source fractions have been updated from 0.1, 0.225, 0.225, 0.225 and 0.225 to values of 0.04, 0.14, 0.19, 0.49, and 0.14 for the five bins. These updated source functions allocate more fraction to coarser size bins, following the suggested Brittle Fragmentation Theory (BFT) as proposed by Kok et al. (2011).'

- Line 139-141: Could you specify which algorithm you are referring to here, exactly? Is this from previous work? If so, please provide a reference.

Reply: We have provided a reference here.
'In this study, we implement the eight minerals from the soil mineralogy map provided by C1999 in GFDL AM4.0 to resolve dust mineralogy. The soil map is based on soil analyses that are usually done after wet sieving, which fully disperse mineral aggregates into small particles. This dispersal is particularly relevant for the phyllosilicates, typically found in the form of aggregates in soils. They are detected in the atmosphere at higher proportions at coarser (silt) sizes than those reported in the soil maps (Perlwitz et al., 2015; Perez Garcia-Pando et al., 2016). These recent studies also show that the Brittle Fragmentation Theory (BFT; Kok, 2011) represents a practical framework to generate the emitted particle size distribution based on the dispersed soil PSD, which facilitate the utilization of soil mineralogy maps. In our simulations, we employ BFT to reconstruct the mineral aggregates emitted from the original undispersed soils, following the methods described in Gonçalves Ageitos et al., 2023.'

- Line 154-6: This 5% seems arbitrary. Could you elaborate on how you chose this? What is the sensitivity of your results to this external mixing threshold?

Reply:  Thanks for the comment.
The choice of 5% mass percentage for iron oxides follows the suggestion in Perlwitz et al. (2015a). This number is based on a few measurements. It is expected that EMIT data (Green et al., 2020) will provide a more precise choice of this parameter. This threshold has also been adopted in Gonçalves Ageitos et al. (2023). We have modified the corresponding part of the manuscript as:

All minerals are considered to be externally mixed, except for iron oxides. A large part of the emitted flux of iron oxides is considered to be internally mixed with other minerals, e.g., in the form of accretions in phyllosilicates, in line with observational evidence and previous modeling studies (Kandler et al., 2009; Perlwitz et al., 2015a; Zhang et al., 2015; Panta et al., 2023). As suggested by Gonçalves Ageitos et al. (2023), we define two different types of tracers for the iron oxides: one set of tracers carries the mass of the hematite that constitutes small accretions in clay minerals (i.e., internally mixed with clay minerals), are allowed to be up to 5 % of the masses of their host minerals at emission (Perlwitz et al., 2015a; Gonçalves Ageitos et al., 2023). Given the low fractional mass of hematite compared to their host minerals, we assume that these accretions do not change the density of their host particles. These internally mixed accretions form the largest fraction of the emitted hematite. Another smaller set of tracers carries the mass of the remaining fraction of hematite, which is considered to be externally mixed with the other minerals, including the internal mixtures of hematite with clay.

- Line 167-170: More details here would benefit the reader and make it easier to appreciate your results. Could you give a basic description of the Maxwell-Garnett and Bruggeman mixing rules and explain the difference in the resulting optical properties?

Reply: Thanks for the great suggestion. We have included a brief description of the three mixing rules in the manuscript as follows.
'The optical properties of the internal mixture of hematite and clay433 are calculated using three mixing rules: volume weighted average (VOL-mixing), Maxwell Garnett mixing rule (MG-mixing) and Bruggeman mixing rule (BM-mixing). Generally, VOL-mixing is used for a quasi-homogeneous mixture, that is when the components have similar refractive index. For cases involving dominant homogeneous host with small inclusions of contrasting composition, MG-mixing is appropriate. BM-mixing is suitable for mixtures that the inclusions virtually occupy the entire volume of the particle, and the host disappears. The detailed discussion regarding the three mixing rules and their applications can be found in (Markel, 2016; Liu and Daum, 2008) . The appropriate selection of mixing rules is important for the determination of the optical properties of the mixtures. Therefore, we incorporate all three mixing rules in this study.'

- Section 2.3 and 2.4: why use AERONET and laboratory SSA, but not satellite data and in situ measurements?

Reply: Thanks for the comment.
In this study, we aim to evaluate modeled SSA of dust aerosols. In the AERONET version 3 level 2 inversion product, the SSA uncertainty is around 0.03 (Sinyuk et al., 2020). Moreover, there is a way (see section 4 in the manuscript) for selecting events dominated by dust over the 21 years of AERONET observations, which allows deriving SSA for dust aerosols. The selected AERONET dust SSA dataset has been used in several previous studies (Gonçalves Ageitos et al., 2023 and Obiso et al., 2023). Independent laboratory measurements performed over resuspended soil samples are also used as an additional verification method. The limitations of this comparison have been discussed in the manuscript. Both AERONET-based and laboratory SSA observations are considered as good proxies for the dust SSA.
However, satellite or in situ measurements measure total aerosol. Additionally, satellite SSA retrievals are subject to large uncertainties. For example, POLDER GRASP SSA has a correlation of R=0.536 and RMSE=0.0536 compared to AERONET L2 SSA. Similarly, in-situ SSA measurements are also limited by several factors: 1) The extent to which the coarse mode

is measured. Instruments often face difficulties in sampling super-micrometer particles due to inlet inefficiencies. For example, the cutoff diameter of particles sampled by nephelometer is around 2.5 μm in Ryder et al. (2013, 2018) and is 12 μm in Denjean et al. (2016). The sampling loss of coarse particles leads to an overestimation of the in-situ measured dust SSA. 2) Contamination from other aerosols. Aircraft measurements over Mediterranean during June-July 2013 (Denjean et al., 2016) and over West Africa during June-July 2016 (Denjean et al., 2020) are susceptible to pollution contamination. 3) The limited spatial coverage of in situ measurements in each campaign. In situ measurements from different campaigns are subject to different limitations, such as different cut-off diameter and process to deal with contamination by other aerosols. Therefore, the large discrepancy in these measurements may either amplify or eliminate the spatial variation in dust SSA. Given the aforementioned limitations in satellite and in situ measurements, we chose not to compare our modeled SSA with them.

    Table 2: which mixing rule was used for HD09 and HD27?

Reply: This is clarified in section 2.1 as follows.
'The dust RI in the standard AM4.0 is taken from Balkanski et al., (2007), assuming a fixed hematite content of 2.7% by volume (HD27), which is calculated for the internal mixture of hematite and five other minerals (calcite, quartz, illite, kaolinite and montmorillonite) using the Maxwell Garnett mixing rule. The control run conducted with the standard AM4.0 model is labeled as HD27 as described in Table 2.
In addition, we conduct simulations assuming homogeneous dust with hematite content of 0.9% by volume, with RI from Balkanski et al. (2007).'

   • Line 252-3: What data are you using for the t-test? Would this be the result of the 19 individual years in each run? Please elaborate.

Reply: Yes, two groups of simulations are used for the t-test, each group consists of 19 members of one-year simulation. We revised the manuscript to provide further elaboration on this point as follows.
'In this study, for each contrasting pair, we define the anomaly as the group mean difference (based on 19-year mean) between mineral-resolved experiment and control run. An anomaly is considered statistically significant if the p-value, determined by the student's t-test between the two contrasting groups of simulations, is smaller than 0.05.'

   • Figure 1: The imaginary index of refraction here makes no sense to me here. This should decrease with wavelength in the SW spectrum for hematite, HD09, and HD27 as dust is well-known to be more absorbing in blue and UV than red (e.g., Di Biagio et al., ACP, 2019). Is there a mistake in this plot? Also, the dotted line should be defined in the caption and the black and gray lines are invisible in panel a.

Reply: Thanks for the comment.
We have double checked the data used to generate the plot (Scanza et al. 2015), there is no mistake in the plot. I have modified the plot to improve the clearance. For example, we have made black and grey lines more visible in the plot. We have added the definition of dotted lines in the legend. We have changed the scale of x-axis to make the LW spectrum more visible.

[Figure]

- Line 270: I think "solar spectrum" here should be "visible band" instead, based on the caption to Table 3. Also, please indicate the effect of averaging over the visible band, so your results can be compared with those of other studies, which usually report AOD at 500 or 550 nm.

Reply: Thanks for pointing this out.
We have revised this sentence and added one more sentence to explain what the solar spectrum refers to here. The revised part is as follows: 'Additionally, the spectral-averaged DAOD and SSA are always weighted by the solar radiation intensity of the corresponding wavelengths, peaking around 0.50 μm, as shown in Eq. (1) and Eq. (2).

$$\overline{DAOD} = \frac{\int_{\lambda 1}^{\lambda 2} DAOD(\lambda)\, F(\lambda)\, d\lambda}{\int_{\lambda 1}^{\lambda 2} F(\lambda)\, d\lambda}$$   Eq. (1)

$$\overline{SSA} = \frac{\int_{\lambda 1}^{\lambda 2} SSA(\lambda)\, DAOD(\lambda)\, F(\lambda)\, d\lambda}{\int_{\lambda 1}^{\lambda 2} F(\lambda)\, DAOD(\lambda)\, d\lambda}$$   Eq. (2)

Where $F(\lambda)$ describes the solar radiation energy intensity, which can be calculated by Planck's function B(T,λ) with the temperature of the sun (T = 5800 K).'

There is a reason for us to average over the visible band instead of using one single wavelength (e.g., 500nm and 550nm). First, AERONET SSA retrievals are available only at 440, 675, 870, 1020nm. Obtaining the SSA at 500 or 550nm requires the interpolation, which may not reflect the reality. Moreover, in our model AM4.0, we simulate the SSA averaged over the visible band rather

than at certain wavelengths. Therefore, we average AERONET-retrieved SSA at 440 and 675nm (note, the average is weighted by solar spectrum and dust AOD as shown in Eq. 1 and 2 in the main text) to compare with modeled SSA in the visible band.

- Table 3 should include more recent comparisons than with Huneeus '11, which is getting pretty old, such as from CMIP5 (Wu, Lin & Liu, ACP, 2020), CMIP6 (Zhao, Ryder & Wilcox, ACP, 2022), DustCOMM (Kok et al., ACP, 2021), AeroCom3 (Gliss et al., ACP, 2021), etc

Reply: Thanks for the great suggestion. More recent comparisons have been added to Table 3.

- Line 337 onward: Please clarify how exactly you calculate the standard deviation on SSA. Is this only for model values at the AERONET stations in Figure 2?

Reply: Thanks for pointing this out.
The standard deviation of SSA is only for model values at the AERONET stations in Figure 2. This has been clarified at the beginning of Section 4 as follows.
'The modeled dust SSA is evaluated against observation-based results utilizing the following evaluation metrics: the mean SSA (mSSA) is calculated based on SSA for all locations displayed in Figure 2; the standard deviation ($\sigma$), derived from SSA for all locations displayed in Figure 2, is used as an indicator of dust SSA spatial variation, the normalized mean bias ($nMB$) and normalized root mean square error ($nRMSE$) are utilized to assess the mean bias and root mean square error, respectively, of modeled SSA in comparison to observed SSA. Definitions of $nMB$ and $nRMSE$ are provided in Section S2 in the Supplement.'

- Line 337 onward: Please include a plot (in the supplement or the main text) of the SSA standard deviation versus SSA showing the various experiments and the AERONET data to inform the discussion here.

Reply: Thanks for the great suggestion.
The following figure has been added in the supplement as Figure S4.

[Figure]

Figure S4. The standard deviation ($\sigma$) of SSA versus the mean SSA (mSSA) at AERONET sites shown in Figure 2 in the main text. The orange line is for homogeneous dust HD27 and HD09. The green line is for mineral-resolved dust VOL, MG, and BM. The purple star represents AERONET retrievals.

- Line 359: I think it's worth emphasizing here that the dynamic range of SSA is greatly underestimated (model variability in SSA is several times smaller than seen in the real world!), even after accounting for mineralogy. There's clearly something important still missing.

Reply: Thanks for the great suggestion. We have revised this part to emphasize the point as follows.
'The significant underestimation of regional SSA contrast in AM4.0, even after accounting for mineralogy, implies that something important is still missing in models. For instance: 1) the observed regional contrast in iron oxides content may be higher than that in the soil map used in this study, and 2) the model may have underestimated regional contrasts in the dust aerosol size distribution and thus their contribution to SSA, and 3) spatial variation of dust shape, which is not taken into account in the model.'

- Line 435: please state here or elsewhere whether your model accounts for LW scattering or just LW absorption.

Reply: Thanks. Our model only accounts for LW absorption. We have included this information

- Table 4: please clarify what the standard deviation represents. Is this the standard deviation over the 19 simulation years?

Reply: Thanks. We clarified this in the table caption. It is the standard deviation over the 19 simulation years.

- Line 593-6 and Section 5.3: for the benefit of the reader, please discuss here the dynamically situation that causes absorption in this instance to increase ascent and precip rather than increase stability and decrease precip, as it does in many other situations and globally (e.g., Samset et al., Comm. Earth & Env., 2022)

Reply: Thanks for the great comment.

Our global precipitation anomalies are actually consistent with the scaling used by Samset (2022). As described by (e.g.) Allen and Ingram (2002), this scaling is based upon a global balance between atmospheric radiative cooling and the transfer of latent and sensible energy from the surface to the atmosphere. Aerosols like dust that heat the atmosphere (offsetting the radiative cooling), allow these surface fluxes to decrease, along with the moisture supply to the atmosphere causing global precipitation to drop. The BM experiment, with its reduced dust absorption compared to HD27, exhibits a global increase of precipitation by 0.017 mm/day that is consistent with the global scaling used by Samset (2022). (Roughly half of the change in dust absorption is compensated by precipitation. A quarter is provided by the sensible heat flux and the rest may be related to approximations used to estimate the dust DRE.) Global precipitation is comparable between the BM and HD09 experiments because global dust absorption is nearly identical by construction.

Samset (2022) does not actually calculate regional precipitation; instead, he calculates the precipitation change implied by local aerosol absorption according to the global scaling, even though this scaling omits regional moisture transports that are locally non-negligible. These anomalous transports are driven by regional contrasts in the aerosol DRE (e.g., see Miller et al. 2014), and generally must be calculated by a model, as we have done here. The influence of dust aerosols on precipitation is generally a regional redistribution.

Our finding that a reduction is absorption causes a reduction of Sahel precipitation has precedents. One of the simplest demonstrations was by Stephens et al. (GRL 2004), who prescribed dust concentration within half of the domain of a cloud resolving model with fixed SST and showed that dust absorption was balanced by local adiabatic cooling associated with ascent and precipitation. They also noted that precipitation decreased in the remainder of the domain. While they did not present a domain-wide moisture budget, total precipitation must have decreased to satisfy the domain-wide radiative absorption introduced by dust. Similar local precipitation increases for fixed ocean temperatures (i.e., the fast response) have been found by Miller et al. (2004), Lau et al. (2009), Jin et al. (2016) and other studies.

- Figure 13: it seems odd to me here to calculate the domain average only for the statistically significant anomalies because (1) then the anomalies mean something different for b versus d and (2) the numbers depend on only very few grid points or none at all for the Sahara in panel d, it seems, which probably should not list any anomaly then since it's undefined.

Reply: Thanks for the comment.

We have revised the values in the figure title to include the domain average of anomaly for all grids within the region of interest.

[Figure]

- Lines 733-735: here and elsewhere in the paper, it would be useful to the reader if you compare your results of the effects of dust absorption on precip in the Sahara and Sahel to those in Balkanski et al. (ACP, 2021).

Reply: Thanks for the suggestion.

Balkanski et al. (2021) mainly focused on investigating the enhancement of precipitation over the Sahel induced by absorbing dust, using a fully coupled model compared to a no-dust condition. Our study uses an atmospheric model with prescribed SST and compares mineralogy-resolved cases with mineralogy non-resolved cases. Contrasts between the two studies result from differences in the dust absorption (related to contrasts in the dust size distribution or assumed dust refractive index) or differences between the slow response computed by Balkanski et al. (2021) or the fast responses that we calculate here. Nevertheless, our findings are qualitatively consistent with theirs, indicating that more absorptive dust enhances precipitation over the Sahel region. Therefore, we have cited this paper to support our results regarding the enhancement of precipitation due to dust absorption.

[revised manuscript text omitted]